# Syngap1 regulates the synaptic drive and membrane excitability of Parvalbumin-positive interneurons in mouse auditory cortex

Ruggiero Francavilla[1,2], Bidisha Chattopadhyaya[1], Jorelle Linda Damo Kamda[1,2], Vidya Jadhav[1,2], Said Kourrich[3,4,5], Jacques L Michaud[1,2,6], Graziella Di Cristo[1,2]*

[1]CHU Sainte-Justine Azrieli Research Centre, Montreal, Canada; [2]Department of Neurosciences, Université de Montréal, Montréal, Canada; [3]Département des sciences biologiques, UQAM, Montreal, Canada; [4]Centre d'Excellence en Recherche sur les Maladies Orphelines-Fondation Courtois, Pavillon des Sciences biologiques, Montréal, Canada; [5]Center for Studies in Behavioral Neurobiology, Concordia University, Montreal, Canada; [6]Department of Pediatrics, Université de Montréal, Montréal, Canada

*For correspondence:
graziella.di.cristo@umontreal.ca

Competing interest: The authors declare that no competing interests exist.

## eLife Assessment

This study provides **valuable** evidence indicating that SynGap1 regulates the synaptic drive and membrane excitability of parvalbumin- and somatostatin-positive interneurons in the auditory cortex. Since haplo-insufficiency of SynGap1 has been linked to intellectual disabilities without a well-defined underlying cause, the central question of this study is timely. The experimental data is **solid**, as in their revisions the authors successfully addressed questions related to changes in thalamocortical presynaptic excitability, the contradiction between spontaneous and mini EPSCs data, and the anatomical analysis of excitatory synapses.

**Abstract** SYNGAP1 haploinsufficiency-related intellectual disability (SYNGAP1-ID) is characterized by moderate to severe ID, generalized epilepsy, autism spectrum disorder, sensory processing dysfunction, and other behavioral abnormalities. While numerous studies have highlighted a role of Syngap1 in cortical excitatory neurons development, recent studies suggest that Syngap1 plays a role in GABAergic inhibitory neuron development as well. However, the molecular pathways by which Syngap1 acts on GABAergic neurons, and whether they are similar or different from the mechanisms underlying its effects in excitatory neurons, are unknown. Here, we examined whether, and how, embryonic-onset *Syngap1* haploinsufficiency restricted to GABAergic interneurons derived from the medial ganglionic eminence (MGE) impacts their synaptic and intrinsic properties in adult primary auditory cortex (A1). We found that *Syngap1* haploinsufficiency significantly affected the intrinsic properties, overall leading to increased firing threshold and decreased excitatory synaptic drive in Parvalbumin (PV)+ neurons in adult layer IV A1. Further, the AMPA component of thalamocortical evoked EPSC was decreased in PV+ cells from mutant mice. Mutant somatostatin (SST)+ interneurons exhibited decreased spontaneous excitatory input and impaired evoked firing without alterations in firing threshold. Finally, we found that the selective blocking of voltage-gated D-type K+ currents was sufficient to rescue PV+ mutant cell-intrinsic properties to wild-type levels. Together,

these data suggest that *Syngap1* plays a specific role in the maturation of PV+ cell-intrinsic properties and synaptic drive, and its haploinsufficiency may lead to reduced PV cell recruitment in the adult A1, which could in turn contribute to the auditory processing alterations found in SYNGAP1-ID preclinical models and patients.

## Introduction

SYNGAP1 is a key synaptic GTPase-activating protein (GAP) essential for synaptic plasticity, learning, memory, and cognition (*Kim et al., 1998*; *Gamache et al., 2020*). Its expression is abundant within forebrain structures, including the cortex and hippocampus (*Kim et al., 1998*; *Porter et al., 2005*), where it peaks during critical periods of synaptogenesis (*Porter et al., 2005*; *McMahon et al., 2012*; *Gou et al., 2020*; *Jadhav et al., 2024*). *SYNGAP1* is increasingly recognized as a candidate gene in neurodevelopmental disorders, with haploinsufficiency leading to intellectual disability (*SYNGAP1*-ID), epilepsy, autism spectrum disorder, sensory processing deficits, including in the auditory domain, and other behavioral abnormalities (*Hamdan et al., 2009*; *Berryer et al., 2013*; *Carreño-Muñoz et al., 2022*). The role of *Syngap1* has been studied primarily in excitatory neurons. Specifically, *Syngap1* haploinsufficiency has been shown to increase AMPA receptor density and accelerate the maturation of excitatory synapses in hippocampal and somatosensory layer 5 pyramidal cells in rodents (*Clement et al., 2012*; *Clement et al., 2013*; *Ozkan et al., 2014*; *Aceti et al., 2015*). Similarly, xenotransplantation experiments have shown that SYNGAP1-deficient human cortical neurons transplanted into mouse brains exhibit faster synapse formation and maturation, along with disrupted synaptic plasticity (*Vermaercke et al., 2024*). In addition to its synaptic roles, Syngap1 is also implicated in the regulation of cortical neurogenesis of projecting neurons (*Birtele et al., 2023*).

Although Syngap1 research has predominantly focused on excitatory neurons, its mRNA and protein are also expressed in inhibitory neurons (*Zhang et al., 1999*; *Moon et al., 2008*; *Berryer et al., 2016*; *Su et al., 2019*; *Velmeshev et al., 2019*; *Zhao and Kwon, 2023*; *Jadhav et al., 2024*). Emerging evidence suggests a role for *Syngap1* in the migration of GABAergic cells and the maturation of inhibitory synapses (*Berryer et al., 2016*; *Su et al., 2019*; *Sullivan et al., 2020*; *Khlaifia et al., 2023*); nevertheless, whether the underlying cellular and molecular mechanisms are similar to those observed in excitatory cells is unknown.

Cortical inhibitory neurons can be broadly classified into two major subtypes based on their anatomy, physiology, and expression of specific markers: Parvalbumin- (PV+) and somatostatin- (SST+) expressing interneurons (*Rudy et al., 2011*). These subtypes provide perisomatic and distal dendritic inhibition to pyramidal cells, respectively (*Levy and Reyes, 2012*; *Yavorska and Wehr, 2016*). Due to the specificity in their synapse location and distinct functional properties, SST+ and PV+ neurons differentially shape excitatory neuronal responses. By providing fast inhibition onto postsynaptic pyramidal cell somata and proximal dendrites, PV+ cells exert fine control on their output (*Moore and Wehr, 2013*; *Tremblay et al., 2016*), while SST+ cells targeting apical dendrites of postsynaptic pyramidal cells exert specific control over dendritic synaptic integration (*Kawaguchi and Kubota, 1997*; *Chiu et al., 2013*). We recently showed that prenatal-onset *Syngap1* haploinsufficiency restricted to Nkx2.1-expressing GABAergic interneuron precursors, which include PV+ and SST+ interneurons, leads to the development of alterations in auditory cortex activity (*Jadhav et al., 2024*), which resemble those observed in global *Syngap1* haploinsufficient mouse models and *SYNGAP1*-ID patients (*Carreño-Muñoz et al., 2022*), suggesting that interneuron dysfunction may contribute to these specific phenotypes. However, how prenatal-onset *Syngap1* haploinsufficiency in GABAergic interneurons alters their physiology in adult cortex is unknown.

To address this question, we generated conditional transgenic mice wherein *Syngap1* haploinsufficiency was restricted to MGE-derived interneurons. We then assessed the synaptic and intrinsic properties of PV+ fast spiking (FS) and SST+ regular spiking cells in layer IV of the adult primary auditory cortex (A1). We found that both mutant PV+ and SST+ cells show decreased excitatory synaptic drive. Notably, PV+, but not SST+, interneurons showed a significantly increased threshold for action potential (AP) generation, pointing toward a reduced recruitment of cortical PV cells in the mutant mouse. Further, we were able to partially restore PV+ cell function ex vivo using alpha-dendrotoxin (α-DTX), a selective blocker of Kv1 family voltage-gated D-type $K^+$ currents. Overall, these results suggest that

Syngap1 can affect neuronal physiological properties by modulating distinct molecular mechanisms in a cell-type-specific manner.

## Results

### Syngap1 haploinsufficiency in MGE-derived interneurons is associated with decreased synaptic excitation in PV+ cells

Since Syngap1 has been shown to regulate AMPAR-mediated synaptic transmission in hippocampal and cortical excitatory neurons and hippocampal inhibitory neurons (*Ozkan et al., 2014*; *Arora et al., 2022*; *Khlaifia et al., 2023*), we first explored whether embryonic-onset Syngap1 haploinsufficiency in MGE-derived cortical interneurons affect their glutamatergic synaptic inputs in adult A1. We performed targeted voltage-clamp recordings of spontaneous (sEPSCs) and miniature excitatory postsynaptic currents (mEPSCs) in layer IV (LIV) EGFP-expressing interneurons from auditory thalamocortical slices of 9- to 13-week-old Tg(*Nkx2.1-Cre*):*RCE*$^{f/f}$:*Syngap1*$^{+/+}$ (control) and Tg(*Nkx2.1-Cre*):*RCE*$^{f/f}$:*Syngap1*$^{f/+}$ (cHet) littermates (*Figure 1*, *Figure 1—source data 1*, *Figure 1—source data 2*). Nkx2.1-expressing MGE precursors generate most of PV+ and SST+ cortical interneurons (*Xu et al., 2008*), while the RCE allele drives Cre-dependent EGFP expression. Recorded MGE-derived interneurons were filled with biocytin, and their identity was confirmed by immunolabeling for neurochemical markers (PV or SST) and analysis of anatomical properties, including location of the axonal arborization across different cortical layers and presence or absence of dendritic spines (*Figure 1a*, *Figure 1—figure supplement 1*, *Supplementary file 1*, *Figure 1—source data 1*; *Kawaguchi et al., 2006*; *Rock et al., 2018*; *Bertero et al., 2020*). All results were analyzed by linear mixed model (LMM), modeling animal as a random effect and genotype as the fixed effect. This method was chosen for statistical analysis because it accounts for both animals as independent replicates and cell recorded in each mouse as repeated/correlated measures, thus providing the most accurate approach for assessing the data (*Aarts et al., 2014*; *Yu et al., 2022*).

We found that sEPSC amplitude was decreased in LIV PV+ neurons from cHet mice compared to those recorded in control littermates (*Figure 1c* right). sEPSC rise and decay time constants were not significantly different between the two genotypes (*Figure 1—source data 1* and *Figure 1—source data 2*), suggesting that postsynaptic AMPA receptor subunit composition was not affected by Syngap1 haploinsufficiency. To discern whether Syngap1 haploinsufficiency had a pre- or postsynaptic effect on the glutamatergic drive received by PV+ cells, we analyzed mEPSC recorded from the same neurons shown in *Figure 1b–j*. We confirmed that TTX blocked APs in PV+ cells and that recordings were stable as indicated by lack of changes in series resistance during the recording period in our experimental setup (*Figure 1—figure supplement 2f–i*). We found no difference in mEPSC amplitude between the two genotypes (*Figure 1g*, right), indicating that the observed difference in sEPSC amplitude (*Figure 1c*, right) could be due to impaired AP-dependent release in cHet mice and the presence of large-amplitude sEPSCs that are preferentially affected by TTX in control mice (*Figure 1—figure supplement 2b–e*). Conversely, cHet mice showed longer inter-mEPSC time intervals (*Figure 1g*, left, for inter-cell mean comparison LMM, p = 0.056, for cumulative distributions LMM, *p = 0.045), and significantly lower charge transfer and $\Delta Q*f$ (*Figure 1j*) compared to control littermates, suggesting a decrease of glutamatergic presynaptic release sites onto PV+ cells. Recent studies, based on serial block-face scanning electron microscopy, suggest that cortical PV+ interneurons receive more robust excitatory inputs to their perisomatic region than pyramidal neurons (*Hwang et al., 2021*; *Elabbady et al., 2025*). Quantification of the density of putative glutamatergic synapses onto PV+ cell somata, identified by the colocalization of the vesicular glutamate 1 (vGlut1, presynaptic marker) and PSD95 (postsynaptic marker), revealed a significant decrease in cHet compared to control mice (*Figure 1—figure supplement 3a, b*). Whether the density of dendritic targeting excitatory inputs is affected as well remains an open question.

In A1, LIV PV+ cells receive stronger subcortical thalamocortical inputs compared to excitatory cells and other subpopulations of GABAergic interneurons (*Ji et al., 2016*; *Rock et al., 2018*). We thus recorded evoked AMPA (eAMPA)- and NMDA (eNMDA)-mediated currents in PV+ cells by bulk electrical stimulation of the thalamic radiation (*Figure 2a–f*, *Figure 2—source data 1*; *Figure 2—source data 2*), to determine whether thalamocortical synapses were affected by conditional Syngap1 haploinsufficiency. eAMPA amplitude, area under the curve (AUC) charge transfer (average of all responses,

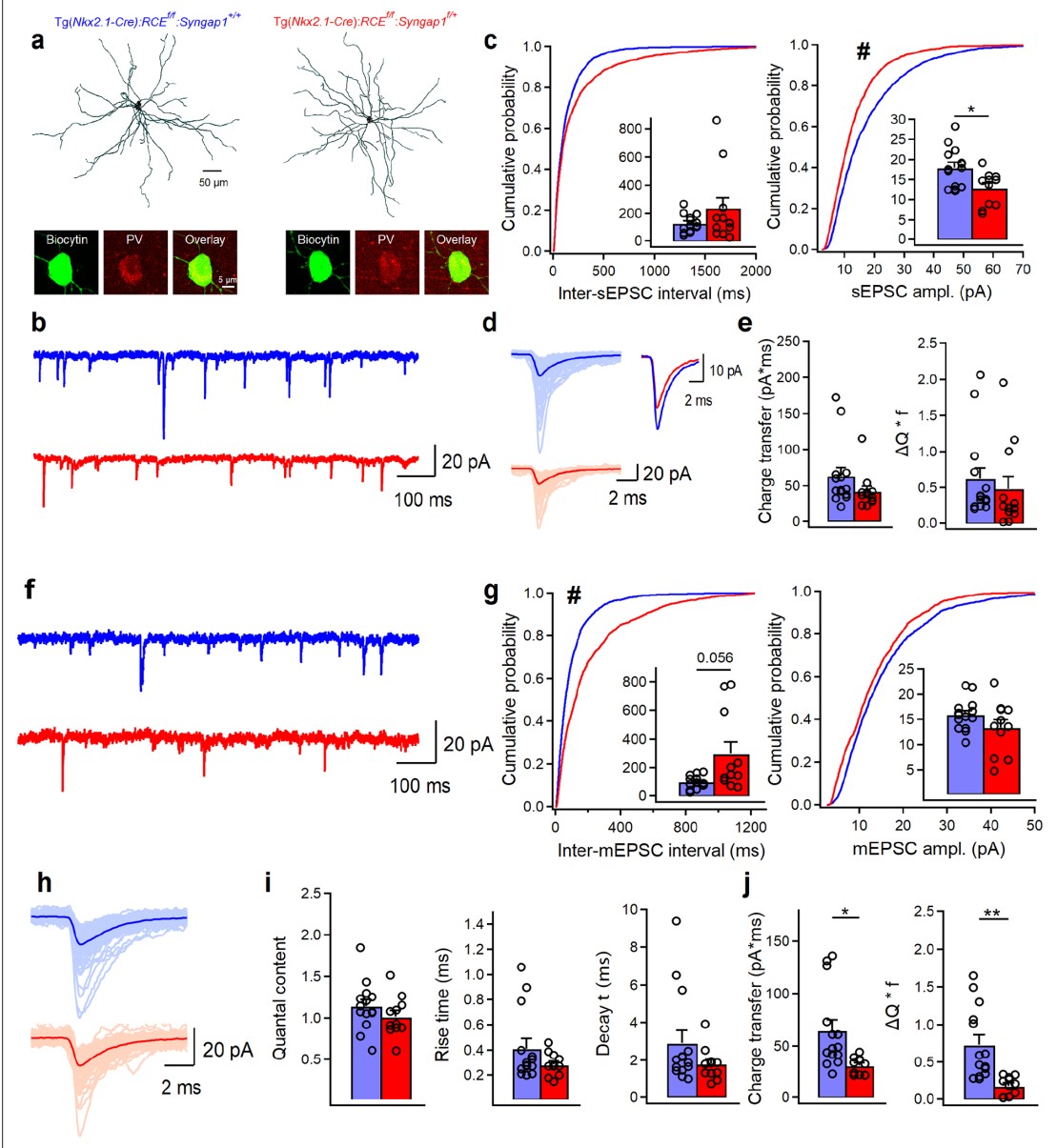

**Figure 1.** *Syngap1* haploinsufficiency in Nkx2.1+ interneurons is associated with reduced sEPSC amplitude and mEPSC frequency in LIV BCs. (**a**) Anatomical reconstructions of PV+ cells filled with biocytin in control (left) and cHet mice (right) during whole-cell patch-clamp recordings and post hoc immunohistochemical validation of BC interneurons confirming the positivity for PV. (**b**) Representative traces of sEPSCs recorded in BC cells from control Tg(*Nkx2.1-Cre*):*RCE^{f/f}*:*Syngap1^{+/+}* (blue, *n* = 14 cells, 7 mice) and cHet Tg(*Nkx2.1Cre*):*RCE^{f/f}*:*Syngap^{f/+}* (red, *n* = 11 cells, 6 mice) mice. (**c**) Cumulative probability plots show a significant decrease in the amplitude of sEPSC in cHet mice compared to control mice (hash sign denotes the significance for LMM related to the cumulative distributions, #p = 0.029) and no change in the inter-sEPSC interval (LMM, p = 0.345). Insets illustrate significant differences in the sEPSC amplitude for inter-cell mean comparison (LMM, *p = 0.014) and no difference for inter-sEPSC interval (LMM, p = 0.230). (**d**) Representative examples of individual sEPSC events (100 pale sweeps) and average traces (bold trace) detected in BCs of control and cHet mice and superimposed scaled traces (right top), (**e**) Summary bar graphs showing no differences for inter-cell mean charge transfer (LMM, p = 0.090, left) and for the charge transfer when frequency of events is considered (LMM, p = 0.140). (**f**) Representative traces of mEPSCs recorded from the same neurons shown in b–e. (**g**) Cumulative probability plots show no change in the amplitude of mEPSC (LMM, p = 0.151) and a significant increase in the inter-mEPSC interval in cHet mice compared to control mice (LMM, #p = 0.045). Insets illustrate summary data showing no significant differences in the amplitude (LMM, p = 0.155) and a trend toward longer inter-mEPSCs intervals for inter-cell mean comparison in cHet compared to control mice (LMM, p = 0.056). (**h**) Representative examples of individual mEPSC events (100 pale sweeps) and average traces (bold trace) detected in BCs of control and cHet mice. (**i**) Summary bar graphs for a group of cells show no significant differences in the quantal content (LMM, p = 0.189) and mEPSCs kinetics (LMM, p = 0.269 for rise time, and p = 0.193 for decay time). (**j**) Summary bar graphs showing a significant decrease in cHet mice for inter-cell mean charge transfer

*Figure 1 continued on next page*

*Figure 1 continued*

(LMM, *p = 0.012, left) and for the charge transfer when frequency of events is considered (LMM, **p = 0.002). Bar graphs represent mean ± SEM. * and # indicates p-value <0.05 for bar graphs and cumulative distribution, respectively.

The online version of this article includes the following source data and figure supplement(s) for figure 1:

**Source data 1.** sEPSCs and mEPSCs in LIV PV+ cells from control vs cHet mice.

**Source data 2.** sEPSCs and mEPSCs in LIV PV+ cells from control vs cHet mice-raw data.

**Figure supplement 1.** Anatomical and neurochemical identification of LIV PV+ and SST+ interneurons in mouse primary auditory cortex.

**Figure supplement 2.** Blockade of voltage-dependent Na⁺ channels by TTX abolished APs in PV+ cells from adult primary auditory cortex.

**Figure supplement 3.** *Syngap1* haploinsufficiency reduces the density of local vGlut1 excitatory inputs without affecting VGlut2 thalamocortical inputs to PV+ cell somata.

successes + failures, *Figure 2b, right, c*) and potency (average of all successes only, *Figure 2e left*) were decreased in cHet mice as compared to control littermates. In addition, we found a substantial increase in onset latencies of eAMPA currents (*Figure 2d*), suggesting a potential deficit in the thalamocortical recruitment of PV+ cells. Next, we assessed eNMDA currents in PV+ cells in the presence of GABA$_A$R, GABA$_B$, and AMPA inhibitors (1 μM Gabazine, 2 μM CGP, and 10 μm NBQX, respectively) (*Figure 2c, e*). We found that eNMDA currents as well as the fraction of PV+ cells showing these responses were similar in cHet and control littermates (*Figure 2c right, 2e left*), therefore leading to increased NMDA/AMPA in cHet mice (*Figure 2e right*). Interestingly, the kinetics of eAMPA and eNMDA currents were similar in both genotypes, indicating no change in their subunit composition (*Figure 2f*, *Figure 2—source data 1*, and *Figure 2—source data 2*). These results suggest that embryonic-onset *Syngap1* haploinsufficiency in MGE-derived interneurons specifically impairs AMPA-mediated thalamocortical recruitment of PV+ cells. Of note, the density of putative thalamocortical glutamatergic synapses onto PV+ cell somata, identified by the colocalization of the vesicular glutamate 2 (vGlut2, thalamocortical presynaptic marker) and PSD95 (postsynaptic marker) was not significantly different in cHet as compared to littermate controls (*Figure 1—figure supplement 3c, d*), suggesting that presynaptic release from excitatory thalamocortical fibers or/and AMPARs expression at thalamocortical synapses on PV+ cells is likely decreased in cHet mice.

To further explore this phenomenon, we performed paired pulse ratio (PPR) experiments, as PPR typically reflects changes in presynaptic release probability (*Figure 2g, h*). We found that, in contrast with Control mice, evoked excitatory inputs to LIV PV+ cells showed paired-pulse facilitation in cHet mice (*Figure 2g, h*), suggesting that thalamocortical presynaptic sites likely have decreased release probability in mutant compared to control mice.

Since PV+ cell recruitment is regulated by the balance of its excitatory and inhibitory inputs, we next analyzed spontaneous (sIPSCs) and miniature inhibitory postsynaptic currents (mIPSCs) recorded from LIV PV+ cells in both genotypes. We observed reduced sIPSC amplitude in cHet compared to control PV+ cells (*Figure 3a, b*, *Figure 3—source data 1*, and *Figure 3—source data 2*); however, mIPSC analysis revealed no genotype-dependent differences in any parameters (*Figure 3f–j*, *Figure 3—source data 1*, and *Figure 3—source data 2*), suggesting that decreased sIPSC amplitude in cHet PV+ cells was likely due to changes in presynaptic cell-intrinsic excitability and/or network activity.

## cHet mice show decreased layer IV PV+ cell-intrinsic excitability

PV+ cell recruitment in cortical circuits is dependent on both their synaptic drive and intrinsic excitability. Thus, we sought to investigate how *Syngap1* haploinsufficiency in MGE-derived interneurons impacts the intrinsic excitability and firing properties of PV+ cells, by performing whole-cell current-clamp recordings (*Figure 4*, *Figure 4—source data 1*, and *Figure 4—source data 2*). We found no changes in passive membrane properties (Cm, Rin, and $\tau$) of PV+ cells recorded from cHet mice as compared to control littermates (*Figure 4a*); conversely, analysis of active membrane properties revealed a significant decrease in the excitability of mutant PV+ cells (*Figure 4b and c*). In particular, cHet PV+ cells showed reduced AP amplitude and increased AP threshold and latency to first AP (*Figure 4*, *Figure 4—source data 1*, and *Figure 4—source data 2*). In line with the decrease in intrinsic excitability, the threshold current was increased in PV+ cells from cHet mice (*Figure 4d*). Both cHet and control PV+ cells displayed typical, sustained high-frequency trains of brief APs with little spike frequency adaptation in response to incremental current injections (*Figure 4e*, right); however,

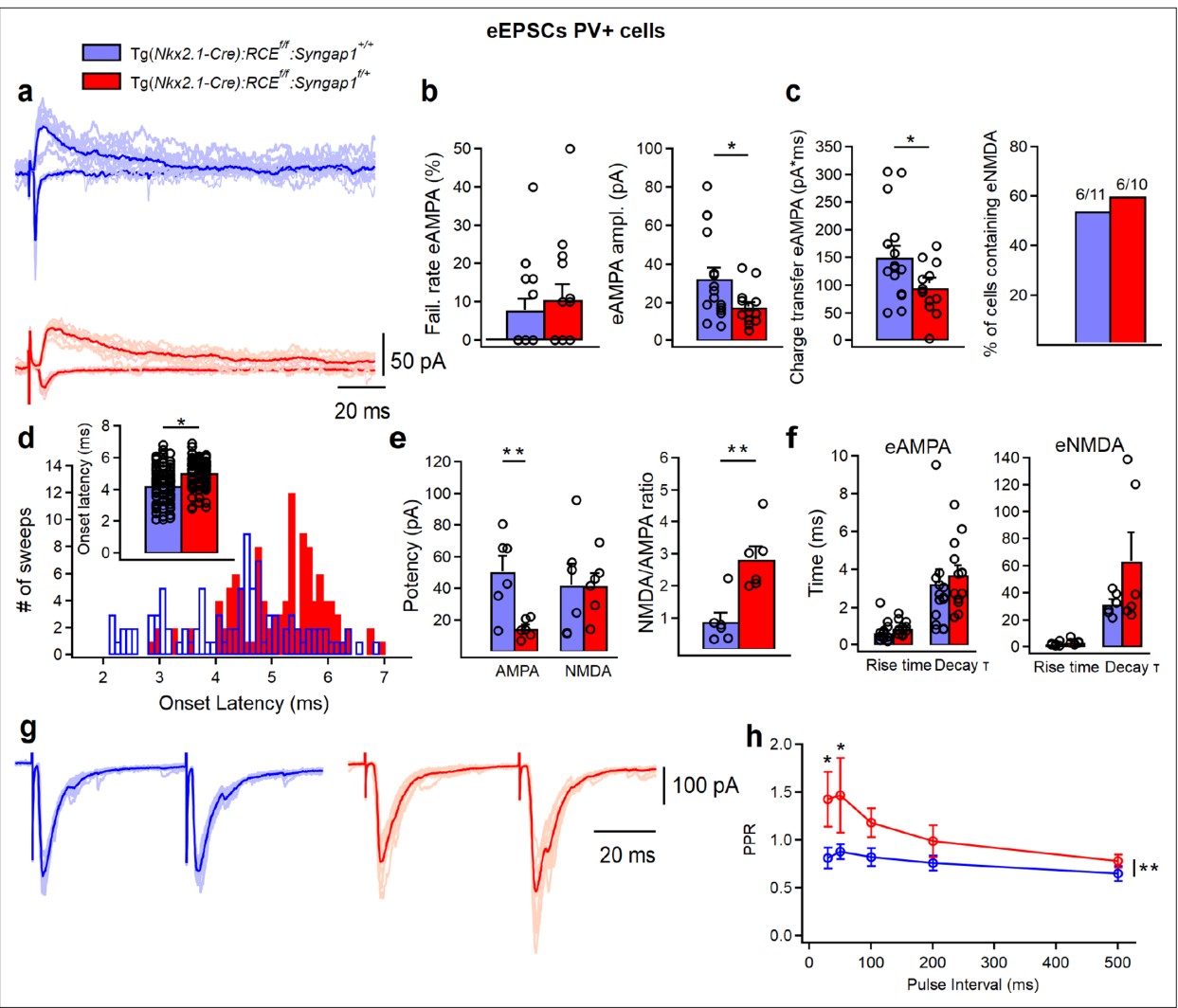

**Figure 2.** Thalamocortical eAMPA transmission is decreased in LIV PV+ cells from Tg(*Nkx2.1-Cre*):*RCE*^f/f^:*Syngap1*^f/+^ mice. (**a**) Representative examples of individual eAMPA (negative deflections) and eNMDA (positive deflections) (5–10 pale sweep) and average traces (bold trace) recorded in PV cells from control (blue, *n* = 16 cells, 7 mice) and cHet mice (red, *n* = 14 cells, 7 mice). (**b**, left) Summary plots showing no change in the failure rate of eAMPA (left, LMM, p = 0.550) and a significant decrease in the minimal (including failures and successes) eAMPA amplitude (**b**, right LMM, *p = 0.031) and (**c**, left) charge transfer (LMM, *p = 0.033) in cHet mice. (**c**, right) Summary bar graph illustrating the percentage of PV+ cells containing eNMDA in the thalamocortical evoked EPSC. (**d**) Synaptic latency histograms (bottom) of thalamocortical eEPSC from control and cHet mice, and summary bar graph (top) illustrating an increase in the onset latencies of eEPSC in cHet mice (LMM, *p = 0.023). For both histograms, bins are 0.1 ms wide. (**e**) Summary plots showing a significant decrease in the potency of eAMPA (successes only, LMM, **p = 0.003, left) in cHet mice with no change in eNMDA (LMM, p = 0.969), A significant increase is present in the NMDA/AMPA ratio (i.e., ratio of the peak for eNMDA and eAMPA, LMM, **p = 0.001, right) in cHet mice. (**f**) Summary plots showing no significant differences in the eAMPA (LMM, p = 0.177 for rise time, and p = 0.608 for decay time) and eNMDA kinetics (LMM, p = 0.228 for rise time, and p = 0.221 for decay time). (**g**) Representative examples of individual LIV evoked EPSC (10 pale sweeps) and average traces (bold trace) with an interval of 50 ms recorded in two BC cells from control (blue) and cHet mice (red). (**h**) Summary plot showing significantly increased PPRs recorded from LIV BC cHet (red circles, *n* = 8 cells; 5 mice) compared to controls (blue circles, *n* = 9 cells, 5 mice), when two EPSCs were evoked in LIV BC with two electric pulses at 30- or 50-ms intervals (two-way repeated measure ANOVA with Sidak's multiple comparison post hoc test, **p = 0.001). Bar graphs represent mean ± SEM. * indicates p-value <0.05; ** indicate p-value <0.005.

The online version of this article includes the following source data for figure 2:

**Source data 1.** eAMPA and eNMDA currents in LIV PV+ cells from control vs cHet mice.

**Source data 2.** eAMPA and eNMDA currents in LIV PV+ cells from control vs cHet mice-raw data.

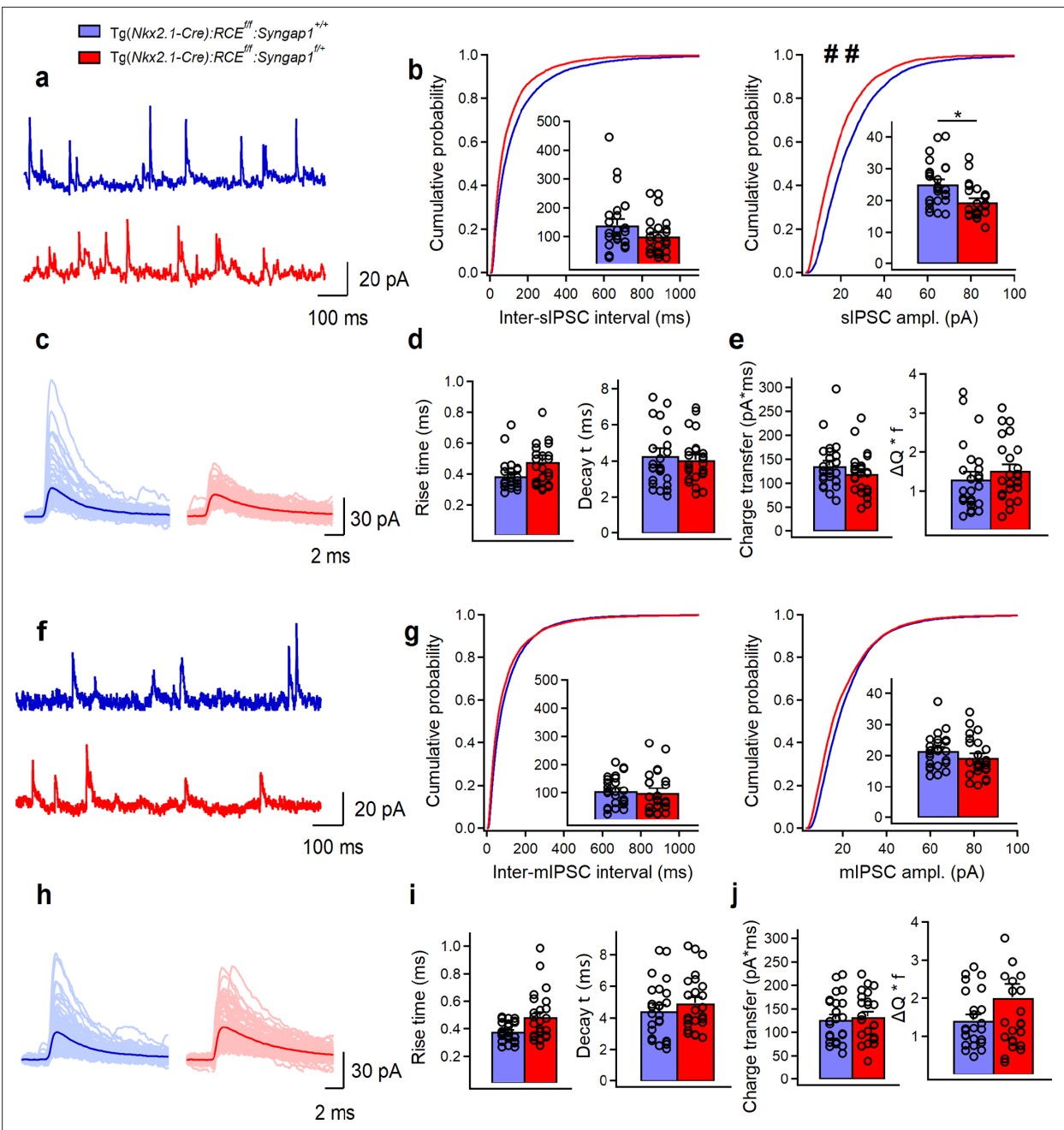

**Figure 3.** The amplitude of sIPSCs, but not mIPSCs, in LIV PV+ cells is reduced in Tg(*Nkx2.1-Cre*):*RCE^f/f*:*Syngap1^f/+* mice. (**a**) Representative traces of sIPSCs recorded in PV+ cells from control (blue, *n* = 25 cells, 8 mice) and cHet (red, *n* = 24 cells, 7 mice) mice. (**b**) Cumulative probability plots show a significant decrease in the amplitude of sIPSC in cHet mice compared to control (LMM, ##p = 0.003) and no change in the inter-sIPSC interval (LMM, p = 0.106). Insets illustrate significant differences in the sIPSC amplitude for inter-cell mean comparison (LMM, *p = 0.009) and no difference for inter-sIPSC interval (LMM, p = 0.185). (**c**) Representative examples of individual sIPSC events (100 pale sweeps) and average traces (bold trace) detected in PV cells of control and cHet mice. (**d**) Summary bar graphs for a group of cells show no significant differences in the sIPSCs kinetics (LMM, p = 0.113 for rise time, and p = 0.602 for decay time). (**e**) Summary bar graphs showing no differences for inter-cell mean charge transfer (LMM, p = 0.234, left) and for the charge transfer when frequency of events is considered (LMM, p = 0.273). (**f**) Representative traces of mIPSCs recorded in PV+ cells from control (blue, *n* = 25 cells, 8 mice) and cHet (red, *n* = 24 cells, 7 mice) mice. (**g**) Cumulative probability plots show no change in the amplitude of mIPSC (LMM, p = 0.118) and in the inter-mIPSC interval (LMM, p = 0.411). Insets illustrate summary data showing no significant differences in the amplitude (LMM, p = 0.195) and the inter-mIPSCs interval for inter-cell mean comparison (LMM, p = 0.243). (**h**) Representative examples of individual mIPSC events (100 pale sweeps) and average traces (bold trace) detected in PV+ cells of control and cHet mice. (**i**) Summary bar graphs for a group of cells show no significant differences in the mIPSCs kinetics (LMM, p = 0.103 for rise time, and p = 0.597 for decay time). (**j**) Summary bar graphs showing no differences for

*Figure 3 continued on next page*

*Figure 3 continued*

inter-cell mean charge transfer (LMM, p = 0.374, left) and for the charge transfer when frequency of events is considered (LMM, p = 0.100). Bar graphs represent mean ± SEM. * indicates p-value <0.05 for bar graphs. ## indicates p-value <0.005 for cumulative distribution.

The online version of this article includes the following source data for figure 3:

**Source data 1.** sIPSCs and mIPSCs in LIV PV+ cells from control vs cHet mice.

**Source data 2.** sIPSCs and mIPSCs in LIV PV+ cells from control vs cHet mice-raw data.

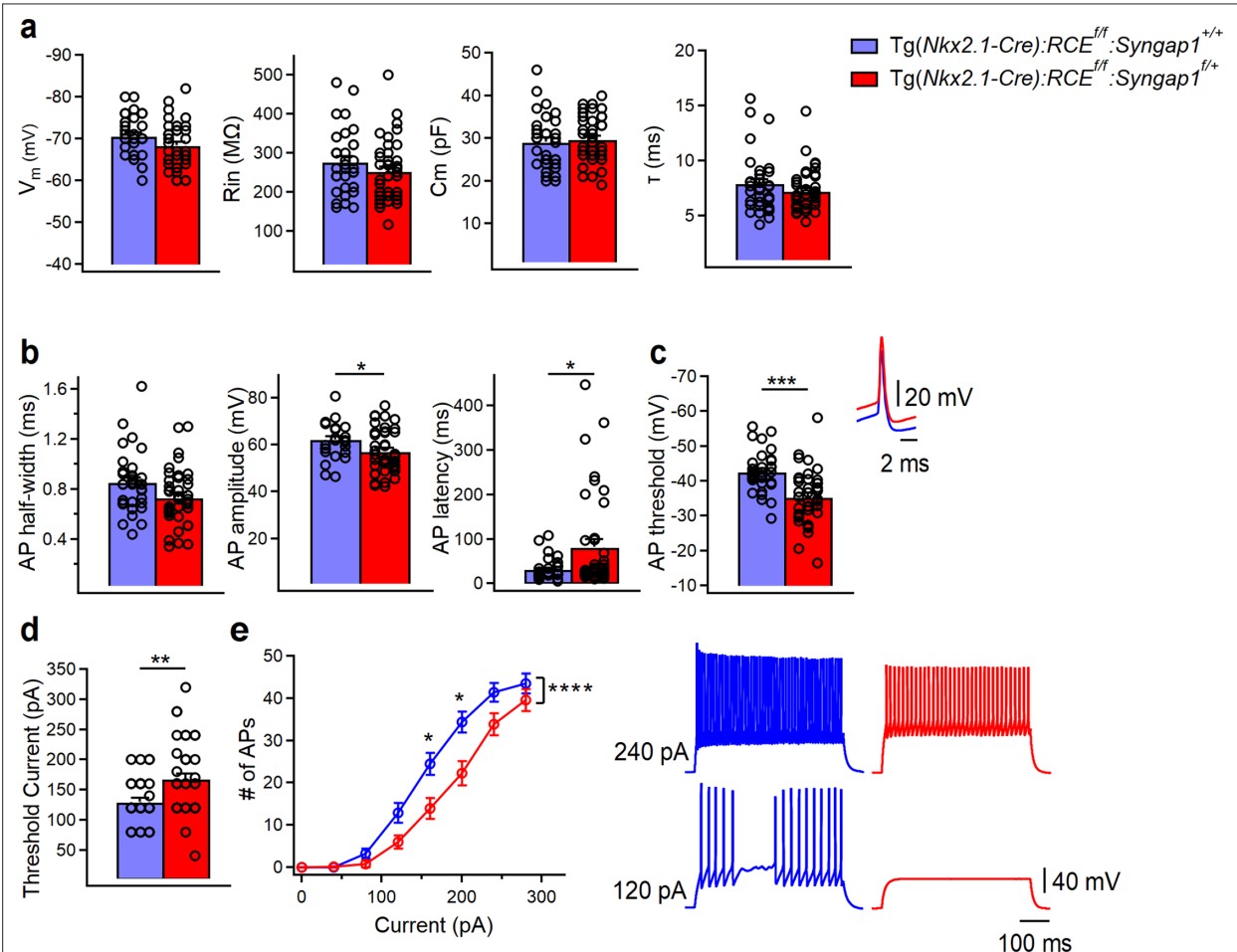

**Figure 4.** PV+ cells intrinsic excitability is decreased in Tg(*Nkx2.1-Cre*):*RCE*<sup>f/f</sup>:*Syngap1*<sup>f/+</sup> mice. (**a**) Summary data showing no changes in the passive membrane properties between control (blue, n = 33 cells, 15 mice) and cHet mice (red, n = 40 cells, 17 mice) (LMM, p = 0.081 for Vm, p = 0.188 for Rin, p = 0.188 for Cm, p = 0.199 for $\tau$ ). (**b**) Summary data showing no differences in AP half-width (LMM, p = 0.111) but a significant decrease in AP amplitude (LMM, *p = 0.032) and a significant increase in AP latency (LMM, *p = 0.009) from PV+ cells recorded in cHet mice. (**c**, left) The summary bar graph shows a significant increase in AP threshold from cHet mice (LMM, ***p < 0.001) for the first AP generated. (**c**, right top) Representative single APs evoked by threshold currents from control and cHet mice. APs are aligned at 50% of the rising phase on the *X* axis and peak on the *Y* axis. Note the more hyperpolarized AP with consequent reduction in AP amplitude in PV+ cells from cHet mice. (**d**) Summary bar graph shows a significant increase in the threshold current (LMM, **p = 0.004). (**e**, left) Summary plot showing a reduction of averaging number of APs per current step (40 pA) amplitude recorded from LIV PV+ cHet (red circles, n = 38 cells; 18 mice) compared to control (blue circles, n = 30 cells, 15 mice) neurons (two-way repeated Measure ANOVA with Sidak's multiple comparison post hoc test, ****p < 0.0001). (**e**, right) Representative voltage responses indicating the typical FS firing pattern of PV+ cells in control and cHet mice in response to depolarizing (+120 and +240 pA) current injections corresponding to threshold current and 2× threshold current. Bar graphs represent mean ± SEM. * indicates p-value <0.05; *** indicates p-value <0.001; **** indicates p-value <0.0001.

The online version of this article includes the following source data for figure 4:

**Source data 1.** Membrane properties of total LIV PV+ cells population in control vs cHet mice.

**Source data 2.** Membrane properties of total LIV PV+ cells population in control vs cHet mice-raw data.

cHet PV+ interneurons fired significantly fewer APs in response to the same depolarizing current injection when compared to control mice (*Figure 4e*, left; two-way repeated measure ANOVA with Sidak's multiple comparison post hoc test, ****p < 0.0001). These data show that embryonic-onset *Syngap1* haploinsufficiency in PV+ cells impairs their basic intrinsic and firing properties.

## Embryonic-onset *Syngap1* haploinsufficiency in MGE-derived interneurons differentially impacts dendritic arborization and intrinsic properties of distinct PV+ cell subpopulations

The majority of PV+ cells are classified as FS cells, due to their ability to sustain high-frequency discharges of APs (*Figure 4e*, right). However, clusters of atypical PV+ cells have been previously reported in several brain regions including subiculum (*Nassar et al., 2015*), striatum (*Bengtsson Gonzales et al., 2020*), hippocampus (*Ekins et al., 2020*) and somatosensory cortex (*Helm et al., 2013*). While atypical PV+ cells share many electrophysiological parameters with FS cells, they have a slower AP half-width and possess a lower maximal AP firing frequency (*Helm et al., 2013*; *Nassar et al., 2015*; *Bengtsson Gonzales et al., 2020*; *Ekins et al., 2020*). Using these two criteria, we found a moderate negative correlation between $F_{max}$initial and AP half-width in both genotypes (*Figure 5a*, left). This suggests that PV+ cell with broader AP durations may have a lower $F_{max}$initial, a feature previously observed in atypical PV+ cells from other cortical areas (*Helm et al., 2013*; *Nassar et al., 2015*; *Bengtsson Gonzales et al., 2020*; *Ekins et al., 2020*). To further investigate whether A1 PV+ cells from control mice could be functionally segregated into distinct clusters, we performed hierarchical clustering of AP half-width and $F_{max}$initial values based on Euclidean distance (*Figure 5a*, right). This analysis identified two clusters of PV+ cells: one with short AP half-widths associated with higher $F_{max}$initial values, and another with broader AP durations and lower $F_{max}$initial values, consistent with what was reported in other cortical regions (*Helm et al., 2013*; *Nassar et al., 2015*; *Bengtsson Gonzales et al., 2020*; *Ekins et al., 2020*). Although hierarchical clustering distinguished these two subgroups, a few PV+ cells with longer AP half-widths exhibited $F_{max}$initial values typical of PV+ cells with shorter AP half-widths (*Figure 5a*, right), indicating that these two parameters alone may not be sufficient to fully differentiate subtypes within our PV+ cell dataset. We thus decided to perform principal component analysis (PCA) using additional key intrinsic physiological features such as passive ($V_m$, $R_{in}$, and $C_m$) and active (threshold current, AP half-width, AP amplitude, first AP latency, AP threshold, fast afterhyperpolarization [fAHP] amplitude, amplitude AR, frequency AR, $F_{max}$initial, and $F_{ss}$) membrane properties (*Figure 5b*, left). We then selected the intersection point of the two AP half-width distributions in control and cHet mice as a cut-off to define two different subpopulations of PV+ cells: basket cell (BC)-short (AP half-width <0.78 ms) and BC-broad (AP half-width ≥0.78 ms) (*Figure 5c, d*). In control mice, these two PV+ cell subtypes showed major differences in $R_{in}$, $C_m$, $F_{max}$initial, and $F_{ss}$ (*Source data 1*), comparison between BC-short and BC-broad in control mice. PCA analysis also revealed that, while two distinct PV+ cell subgroups were clearly distinguishable in control mice, these differences were more ambiguous in cHet mice, wherein some BC-short cells fell within the BC-broad subgroup (*Figure 5d*). In addition, we observed that the fraction of BC-short cells was increased in cHet compared to control mice (61% vs 41% of total PV+ cells, respectively), suggesting that *Syngap1* haploinsufficiency affects specific subgroups of PV+ cells (*Figure 5d*).

Next, we examined whether the diversity in PV+ cell electrophysiological profiles was reflected in their dendritic arborization (*Figure 5e–h*, *Figure 5—source data 1–8*). We also measured the anatomical location of cell bodies (distance in μm from the pia) to confirm that the recorded and analyzed cells were situated within LIV (*Figure 5g, h*). In control mice, both BC-short and BC-broad cells showed ovoid somata and multipolar dendrites (*Figure 5e, f*). Anatomical reconstruction and morphometric analysis revealed differences in dendritic arborization that correlated positively with AP half-width in control mice (*Figure 5g, k*). In particular, BC-short cells showed significantly lower branch point numbers and dendritic surface area as compared to BC-broad cells (*Figure 5g*). In contrast, the dendritic arbor of BC-short neurons vs BC-broad did not show significant differences in cHet mice (*Figure 5h*). Direct comparison of PV+ cell dendritic arbor in cHet vs control littermates clearly showed that BC-short neurons were specifically affected by *Syngap1* haploinsufficiency (*Figure 5i, j*), with BC-short cells from cHet mice showing a significant increase in dendritic complexity compared to those from control mice (*Figure 5i*). Further, the strong positive correlation of dendritic surface area with AP half-width was present only in PV+ cells from control mice but disappeared in cHet

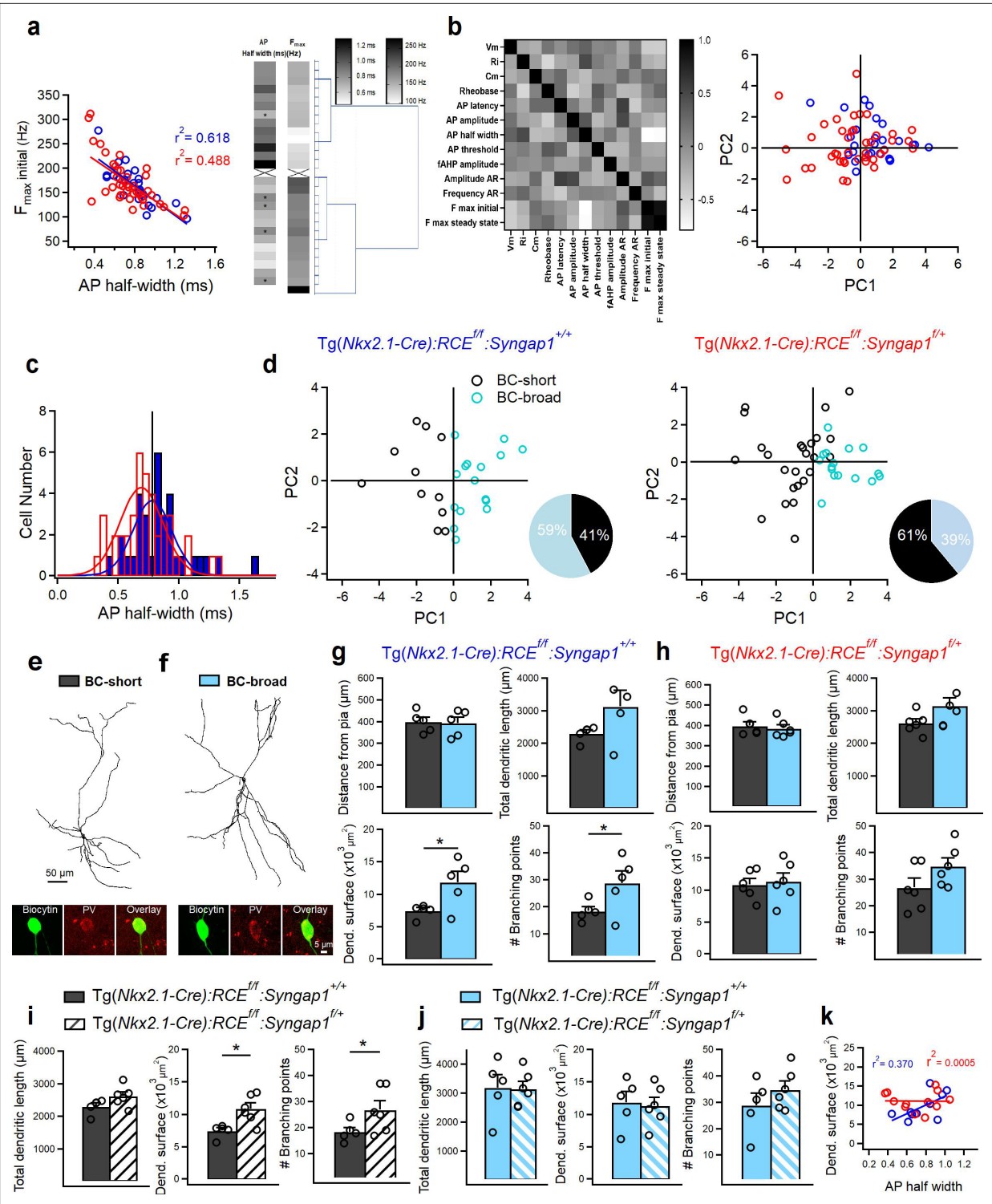

**Figure 5.** *Syngap1* haploinsufficiency in Nkx2.1+ interneurons affects the dendritic arbor of a specific subpopulation of LIV PV+ cells. (**a**, left) Strong negative correlation of $F_{max}$initial with AP half-width in PV+ cells from control (blue, *n* = 33 cells, 15 mice) and cHet mice (red, *n* = 40 cells, 17 mice). (**a**, right) Hierarchical clustering based on Euclidean distance of PV+ cells from control mice. Clustering is based on AP half-width and Max frequency. Asterisks indicate cells with longer AP half-width falling into the cluster including PV+ cells with higher values of $F_{max}$initial. (**b**, left) Correlation of parameters describing membrane properties of PV+ interneurons. The 13 passive and active membrane properties used for principal component analysis (PCA) (derived from 27 PV+ cells from control mice; see Materials and methods) are arrayed against each other in a correlation matrix with the degree of correlation indicated by the shading: white is negatively correlated (correlation index of 0), black is positively correlated (correlation index

*Figure 5 continued on next page*

*Figure 5 continued*

of 1, diagonal squares) and light gray not correlated (correlation index of 0). PCA on the 13 parameters to reduce the dimensionality. (**b**, right) The first (PC1) and second (PC2) PC values derived for each interneuron are plotted against each other. No clear separation of subgroups in the scatterplot of first two PCs is present when genotype is taken into consideration. (**c**) Cumulative histograms of AP half-widths in control (*n* = 33 cells, 15 mice) and cHet mice (*n* = 40 cells, 17 mice) fitted with two Gaussian curves. A vertical line indicates the cutoff value at the intersection between the two curves. For both histograms, bins are 0.05 ms wide. (**d**) PCA analysis using the cutoff value of 0.78 ms and the 13 passive and active membrane properties distinguishes two subgroups of PV+ cells with short (black circles) and broad (turquoise circles) AP-half width duration in both genotypes. Insets illustrate pie charts describing the % of two subgroups of PV+ cells in the control and cHet mice. (**e**) Anatomical reconstructions of a BC-short and (**f**) a BC-broad filled with biocytin in control mice during whole-cell patch-clamp recordings and post hoc immunohistochemical validation for PV. (**g**) Summary data in control mice (gray, BC-short *n* = 5 cells, 4 mice; turquoise, BC-broad *n* = 5 cells, 4 mice) showing no significant difference in terms of distance from pia (p = 0.856, LMM) for both subtypes of PV+ cells analyzed indicating LIV location and significant differences in dendritic parameters between the two subpopulations of PV+ cells (LMM, *p = 0.016 for dendr. surface area, *p = 0.043 for # branching points) and no change in total dendritic length (LMM, p = 0.057). (**h**) Summary data in cHet mice (gray, BC-short *n* = 6 cells, 4 mice; turquoise, BC-broad *n* = 6 cells, 3 mice) showing no significant difference in terms of distance from pia (LMM, p = 0.594) for both subtypes of PV+ cell and all dendritic parameters (LMM, p = 0.062 for total dendritic length, p = 0.731 for dendr. surface area, p = 0.081 for # branching points). (**i**) Summary data showing a significant increase in dendritic complexity between control (gray, *n* = 5 cells, 4 mice) and cHet (white, *n* = 6 cells, 4 mice) for the subpopulation of BC-short (LMM, *p = 0.009 for dendr. surface area, *p = 0.048 for # branching points) and no difference for the total dendritic length (LMM, p = 0.070). (**j**) Summary data showing preserved dendritic parameters in cHet (turquoise filled with pattern, *n* = 6 cells, 3 mice) vs control (turquoise, *n* = 5 cells, 4 mice) (LMM, p = 0.967 for total dendritic length, p = 0.784 for dendr. surface area, p = 0.290 for # branching points). (**k**) The strong positive correlation of dendritic surface area with AP half-width is present only in PV+ cells from control mice (blue, *n* = 10 cells, 8 mice) and disappears in cHet mice (red, 12 cells, 7 mice). Bar graphs represent mean ± SEM. * indicates p-value <0.05.

The online version of this article includes the following source data for figure 5:

**Source data 1.** Morphological properties of BC-short vs BC-broad in control mice.

**Source data 2.** Morphological properties of BC-short vs BC-broad in control mice-raw data.

**Source data 3.** Morphological properties of BC-short vs BC-broad in cHet mice.

**Source data 4.** Morphological properties of BC-short vs BC-broad in cHet mice-raw data.

**Source data 5.** Morphological properties of BC-short in control vs cHet mice.

**Source data 6.** Morphological properties of BC-short in control vs cHet mice-raw data.

**Source data 7.** Morphological properties of BC-broad in control vs cHet mice.

**Source data 8.** Morphological properties of BC-broad in control vs cHet mice-raw data.

mice (*Figure 5k*). Altogether, these data indicate that embryonic-onset *Syngap1* haploinsufficiency in Nkx2.1+ interneurons alters dendritic development in a specific subpopulation of PV+ cells, leading to the increased dendritic area and complexity. These structural changes may, in turn, affect their intrinsic excitability and the dendritic integration of synaptic inputs.

Based on the observed heterogeneity in morpho-electric parameters of PV+ cells, we next sought to investigate the effect of *Syngap1* haploinsufficiency on the intrinsic excitability of specific PV+ cell subtypes (*Figure 6*, *Figure 6—source data 1–4*). We found that BC-short cells showed preserved passive membrane properties (*Figure 6a*) but altered active membrane properties (*Figure 6b–e*) in cHet compared to control mice. In particular, we found increased AP threshold affecting AP amplitude (*Figure 6b, c*) and increased threshold current (*Figure 6d*), indicating a decrease in the excitability of cHet BC-short cells. cHet BC-short interneurons displayed AP firing patterns similar to those in control BC-short (*Figure 6e*, right), but fired less APs in response to somatic depolarization (*Figure 6e*, left, two-way repeated measure ANOVA with Sidak's multiple comparison post hoc test, ****p < 0.0001). In contrast, BC-broad neurons had a more hyperpolarized RMP (*Figure 6f*) and increased AP latency and threshold (*Figure 6g, h*) in cHet mice compared to controls. However, in cHet BC-broad neurons, these changes were not translated into decreased ability to generate spikes (*Figure 6i, j*, two-way repeated measure ANOVA with Sidak's multiple comparison post hoc test, p = 0.333). Altogether, these data suggest that BC-short neurons may be overall more vulnerable to *Syngap1* haploinsufficiency than BC-broad neurons. They further indicate that *Syngap1* levels appear to play a common role in determining the threshold for AP generation in all adult PV+ cells.

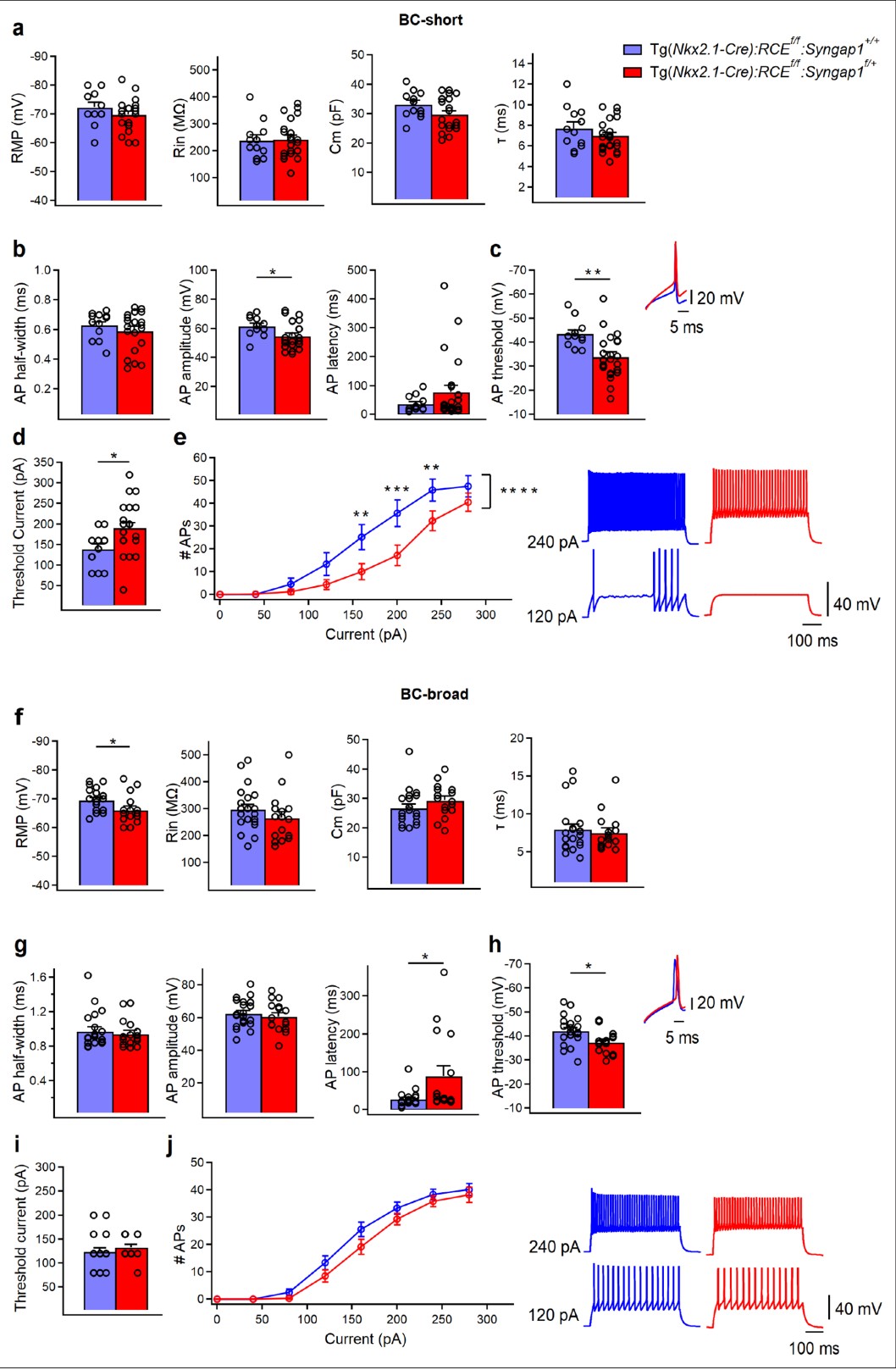

**Figure 6.** Intrinsic excitability is decreased in both subpopulations of PV+ cells in cHet mice. (**a**) Summary data showing no changes in the passive membrane properties of BC-short between control (blue, *n* = 12 cells, 9 mice) and cHet mice (red, *n* = 24 cells, 13 mice) (LMM, p = 0.189 for Vm, p = 0.856 for Rin, p = 0.188 for Cm, p = 0.077 for τ). (**b**) Summary data showing no differences in AP half-width (p = 0.386, LMM) and AP latency (LMM, p =

*Figure 6 continued on next page*

*Figure 6 continued*

0.210) but a significant decrease in AP amplitude (LMM, *p = 0.024) of BC-short recorded in cHet mice. (**c**, left) Summary bar graph shows a significant increase in AP threshold from cHet mice (LMM, **p = 0.002). (**c**, right) Representative single APs evoked by threshold currents from control and cHet mice. (**d**) Summary bar graph shows a significant increase in the threshold current (LMM, *p = 0.015). (**e**, left) Summary plot showing a reduction of averaging number of APs per current step (40 pA) amplitude recorded from LIV BC-short in cHet (red circles, *n* = 22 cells; 13 mice) compared to control (blue circles, *n* = 11 cells, 9 mice) neurons (two-way repeated measure ANOVA with Sidak's multiple comparison post hoc test, ****p < 0.0001). (**e**, right) Representative voltage responses indicating the typical FS firing pattern of BC-short in control and cHet mice in response to depolarizing (+120 and +240 pA) current injections corresponding to threshold current and 2× threshold current. (**f**) Summary data showing a significant decrease in RMP (LMM, *p = 0.023) of BC-broad from cHet mice (red, *n* = 16 cells, 11 mice) but no changes in the other passive membrane properties compared to control mice (blue, *n* = 21 cells, 12 mice) (LMM, p = 0.244 for Rin, p = 0.170 for Cm, p = 0.639 for $\tau$). (**g**) Summary data showing no differences in AP half-width (LMM, p = 0.593) and AP amplitude (LMM, p = 0.713) and a significant increase in AP latency (LMM, *p = 0.035) from BC-broad cells recorded in cHet mice. (**h**, left) Summary bar graph shows a significant increase in AP threshold from cHet mice (LMM, *p = 0.010). (**h**, right) Representative single APs evoked by threshold currents from control and cHet mice. (**i**) Summary bar graph shows no difference in the threshold current (LMM, p = 0.402). (**j**, left) Summary plot showing no difference in the averaging number of APs per current step (40 pA) amplitude recorded from LIV BC-broad in cHet (red circles, *n* = 16 cells, 11 mice) compared to control (blue circles, *n* = 18 cells; 11 mice) neurons (two-way repeated measure ANOVA with Sidak's multiple comparison post hoc test, p = 0.333). (**j**, right) Representative voltage responses indicating the typical FS firing pattern of BC-broad in control and cHet mice in response to depolarizing (+120 and +240 pA) current injections corresponding to threshold current and 2× threshold current. Bar graphs represent mean ± SEM. * indicates p-value <0.05; ** indicates p-value <0.005, **** indicates p-value <0.0001.

The online version of this article includes the following source data for figure 6:

**Source data 1.** Membrane properties of BC-short in control vs cHet mice.

**Source data 2.** Membrane properties of BC-short in control vs cHet mice-raw data.

**Source data 3.** Membrane properties of BC-broad in control vs cHet mice.

**Source data 4.** Membrane properties of BC-broad in control vs cHet mice-raw data.

## *Syngap1* haploinsufficiency in MGE-derived interneurons affects SST+ interneurons firing and spontaneous excitatory inputs

To investigate whether the observed alterations in the synaptic and intrinsic properties of PV+ interneurons are cell-type-specific, we next examined whether the second major group of Nkx2.1-expressing cortical interneurons, the SST +interneurons, exhibits similar changes. In current-clamp recordings, control SST+ cells displayed a low firing rate and characteristic AP frequency accommodation in response to incremental current injections (*Figure 7a*, *Figure 7—source data 1*, and *Figure 7—source data 2*). Morphologically, SST neurons showed ovoid-shaped somata, multipolar (*Figure 1—figure supplement 1b*, left) or bitufted (*Figure 1—figure supplement 1b*, right) dendritic arbours with spines. Their axon projected into layer I, where it arborized, giving rise to multiple collaterals. The molecular identity of this interneuron subtype was confirmed by immunopositivity for SST +and immunonegativity for PV (*Figure 7b*, *Figure 1—figure supplement 1b*). Interestingly, PCA analysis using the previously mentioned electrophysiological parameters clearly distinguished SST+ neurons from BC-short subtype of PV+ cells but showed an overlap between BC-broad PV+ and SST+ cells (*Figure 7c*). These data indicate that, in mature A1, a subtype of PV+ cells share some electrophysiological features with SST+ cells, indicating the necessity to perform post hoc immunohistochemical validation (*Figure 7b, c*). cHet SST+ cells showed no significant changes in the active or passive membrane properties we analyzed (*Figure 7d, e*). However, their evoked firing properties were affected, with fewer APs in response to the same depolarizing current injection compared to control SST + cells (*Figure 7f*, two-way repeated measure ANOVA with Sidak's multiple comparison post hoc test, ***p < 0.001). Furthermore, we found that sEPSC amplitude and charge transfer were significantly decreased in SST+ neurons from cHet mice compared to control littermates (*Supplementary file 2*; *Source data 2*).

Thus, embryonic-onset *Syngap1* haploinsufficiency in MGE-derived interneurons significantly reduced spontaneous excitatory inputs and evoked firing properties of both PV+ and SST+. However,

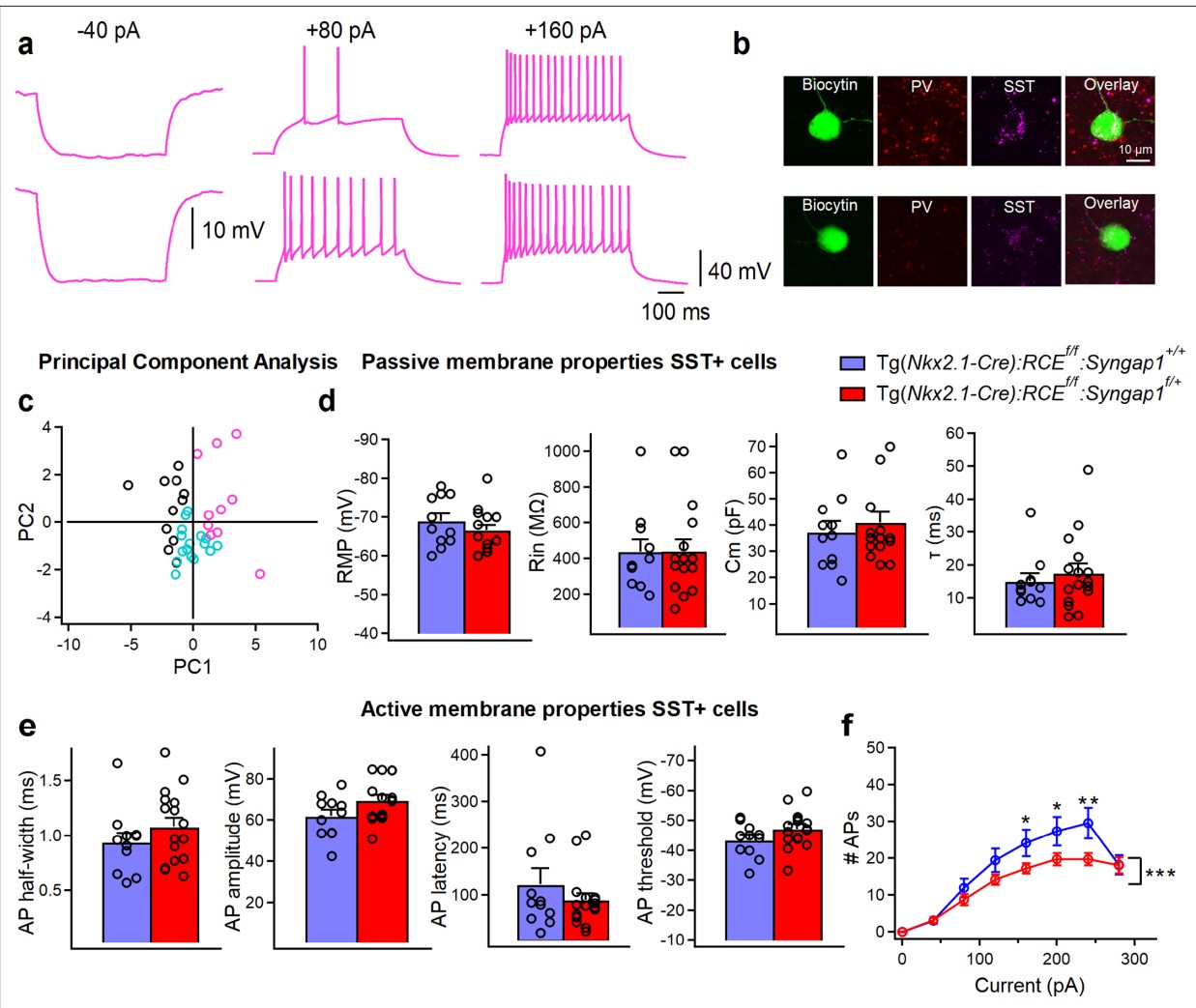

**Figure 7.** Evoked firing properties are reduced in SST+ cells from mice with embryonic-onset *Syngap1* haploinsufficiency in Nkx2.1 interneurons.
(**a**) Representative voltage responses indicating the typical regular adapting firing pattern of SST+ in control mice in response to hyperpolarizing
(−40 pA) and depolarizing (+80 and +160 pA) current injections corresponding to Ih associated voltage rectification, threshold current, and 2× threshold
current, respectively. (**b**) Post hoc immunohistochemical validation of these interneurons confirming their positivity for SST+ and negativity for PV−.
(**c**) Principal component analysis (PCA) using the 13 parameters previously described clearly separates the cluster of SST+ cells (pink circles) from
BC-short (black circles), having however some overlaps with BC-broad (turquoise circles) in control mice. (**d**) Summary data showing no changes in the
passive (LMM, p = 0.283 for Vm, p = 0.959 for Rin, p = 0.484 for Cm, p = 0.501 for $\tau$) and (**e**) active membrane properties (LMM, p = 0.332 for AP half-
width, p = 0.126 for AP amplitude, p = 0.296 for AP latency, p = 0.154 for AP threshold) between SST+ cells from control (blue, n = 11 cells, 8 mice) and
cHet mice (red, n = 16 cells, 9 mice) (**f**) Summary plot showing a reduction of averaging number of APs per current step (40 pA) amplitude recorded from
LIV SST+ in cHet (red circles, n = 16 cells, 9 mice) compared to control (blue circles, n = 11 cells, 8 mice) neurons (two-way repeated Measure ANOVA
with Sidak's multiple comparison *post hoc* test, ***p < 0.001). Bar graphs represent mean ± SEM. * indicates p-value <0.05; ** indicates p-value <0.005.

The online version of this article includes the following source data and figure supplement(s) for figure 7:

**Source data 1.** Membrane properties of total SST+ cells population in control vs cHet mice.

**Source data 2.** Membrane properties of total SST+ cells population in control vs cHet mice-raw data.

**Figure supplement 1.** sEPSC amplitude is reduced in LIV SST+ cells in Tg(*Nkx2.1Cre*):RCE^f/f:*Syngap1*^f/+ mice.

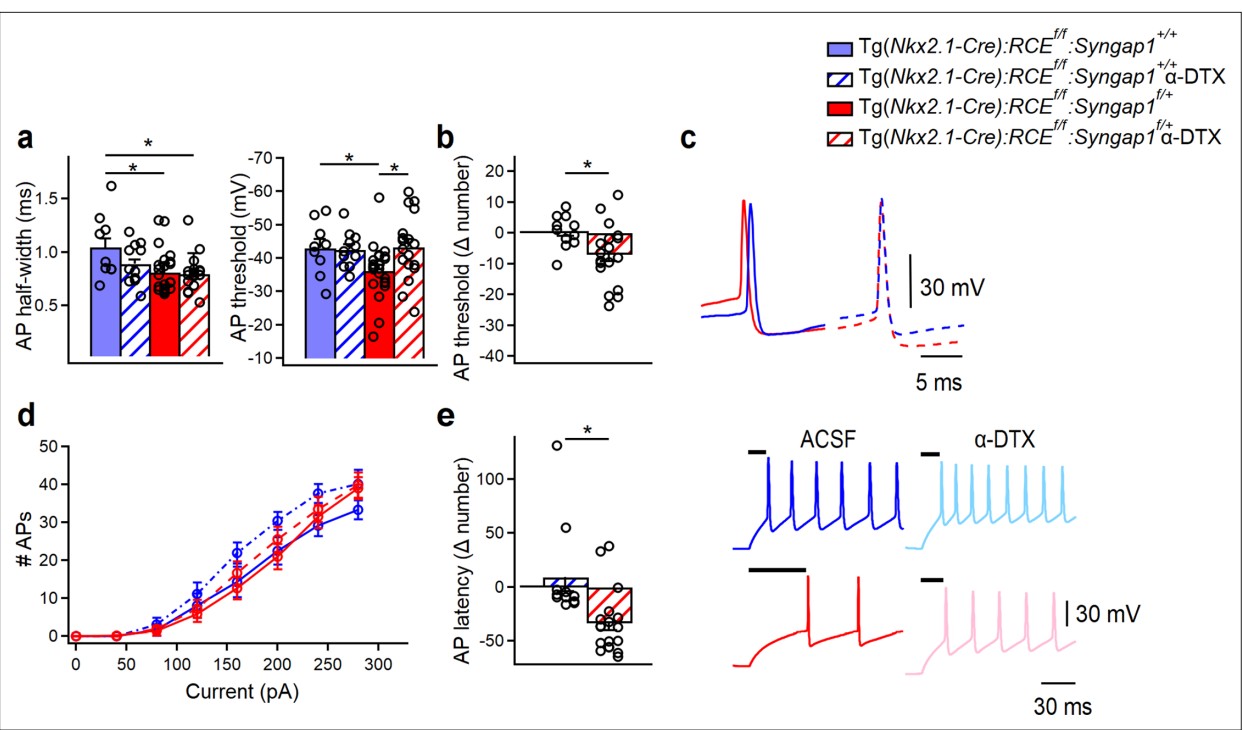

**Figure 8.** *Syngap1* haploinsufficiency alters the intrinsic excitability of LIV PV+ cells by affecting voltage-gated D-type K+ currents. (**a**, left) Summary bar graph shows a significant decrease in AP half-width in PV+ cells from cHet (red) vs control (blue) mice (LMM, *p = 0.034), which persists when cHet PV+ cells are treated with α-DTX (red with diagonal stripes, LMM, *p = 0.039). (**a**, right) Summary bar graph shows a significant increase in AP threshold of PV+ cells from vehicle-treated cHet mice (red) compared to vehicle-treated control mice (blue, LMM, *p = 0.049) and the rescue of this deficit in presence of α-DTX (blue vs red with diagonal stripes, LMM, p = 0.940). (**b**) Delta (Δ) value was calculated for AP threshold by subtracting individual values of α-DTX-treated cells from the average of their respective control group. A significant increase in AP threshold Δ number was found for cHet α-DTX-treated PV+ cells compared to control α-DTX-treated PV+ cells (LMM, *p = 0.015). (**c**) Representative single APs evoked by threshold currents from vehicle-treated control (blue) and cHet (red) mice (center), and control (blue dotted line) and cHet α-DTX-treated (red dotted line) PV+ cells. (**d**) Summary plot showing no difference in the averaging number of APs per current step (40 pA) amplitude recorded from LIV PV+ in cHet and control, both α-DTX-treated and vehicle-treated, PV+ cells (two-way repeated Measure ANOVA with Sidak's multiple comparison post hoc test, p > 0.05). (**e**, left) Summary bar graph shows a significant difference in AP latency Δ number in α-DTX-treated cHet vs α-DTX-treated control PV+ cells (LMM, *p = 0.006). (**e**, right) Representative voltage traces clearly show a reduction in the AP onset for cHet PV+ cells treated with α-DTX (pink trace) compared to vehicle-treated cHet PV+ cells (red trace), while control PV+ cells are not affected (vehicle-treated control PV+ cells, blue traces; control α-DTX PV+ cells, light blue traces). Control mice: vehicle treated, *n* = 9 cells from 4 mice; α-DTX-treated, *n* = 11 cells from 6 mice; cHet mice: vehicle treated, *n* = 23 cells from 10 mice; cHet α-DTX-treated, *n* = 18 cells from 8 mice. Bar graphs represent mean ± SEM. * indicates p-value <0.05.

The online version of this article includes the following source data for figure 8:

**Source data 1.** Membrane properties of PV+ cells with and without α-DTX treatment in control and cHet conditions.

**Source data 2.** Membrane properties of PV+ cells with and without α-DTX treatment in control and cHet conditions-raw data.

alterations in AP initiation, as indicated by an increased firing, were observed exclusively in PV + cells (*Figure 6c and h*), suggesting a selective impairment in their intrinsic excitability.

## A selective Kv1-blocker rescues PV+ cell-intrinsic excitability in cHet mice

Based on previous studies performed in PV+ cells (*Wang et al., 1994*; *Goldberg et al., 2008*; *Zurita et al., 2018*), changes in voltage-gated D-type K+ currents mediated by the Kv1 subfamily could account for the observed altered AP threshold observed in cHet mice. To test this hypothesis, we performed current-clamp recordings from PV+ cells in control and cHet mice in the presence or absence of α-DTX (100 nM), a specific blocker of channels containing Kv1.1, Kv1.2, or Kv1.6 (*Figure 8a–e*, *Figure 8—source data 1*, and *Figure 8—source data 2*). As hypothesized, the presence of α-DTX rescued the voltage threshold for AP generation in cHet PV+ cells to control levels, without affecting the AP shape (*Figure 8a–c*, *Figure 8—source data 1*, and *Figure 8—source data 2*), while the relation between the

AP number and current injection remained the same (*Figure 8d*), indicating that α-DTX had no impact on PV+ cell firing. Further, α-DTX facilitated AP initiation in cHet PV+ cells by reducing AP delay from stimulation onset (*Figure 8e*). These data suggest that *Syngap1* haploinsufficiency may enhance voltage-gated D-type K+ currents, thereby leading to reduced excitability of PV+ cells in the adult A1.

## Discussion

In this study, we investigated the impact of embryonic-onset, MGE-restricted *Syngap1* haploinsufficiency on the intrinsic and synaptic properties of the two major cortical GABAergic interneuron subtypes, PV+ and SST+ cells. In particular, mutant PV+ cells showed reduced intracortical and thalamo-cortical glutamatergic synaptic drive. We further found that *Syngap1* haploinsufficiency has a significant impact on the intrinsic properties, in particular AP threshold, of PV+ cells, resulting in overall decreased excitability. The intrinsic excitability of PV+ cells was rescued in part by pharmacological inhibition of voltage-gated D-type K+ current mediated by the Kv1 subfamily, suggesting that this current may serve as molecular mediator of the functional deficits induced by *Syngap1* haploinsufficiency. Syngap1 has been studied mainly in the context of synaptic physiology; therefore, our data highlights a novel aspect of Syngap1 biology.

Since *Syngap1* mRNA expression in PV+ and SST+ cells is not limited to A1 (*Zhao and Kwon, 2023*; *Jadhav et al., 2024*), it is likely that its haploinsufficiency may affect interneurons physiology in other cortical regions, as well. In our mouse model, Syngap1 haploinsufficiency is driven by the expression of Nkx2.1, which has an embryonic onset (E10.5). However, *Syngap1* expression is thought to be highest during critical periods of synaptogenesis (*Porter et al., 2005*; *McMahon et al., 2012*; *Gou et al., 2020*; *Jadhav et al., 2024*). Therefore, the precise developmental time window during which *Syngap1* insufficiency disrupts PV+ neuron properties remains to be determined.

Syngap1 is thought to be a potent regulator of excitatory synapses, and its reduced expression in excitatory cells causes an increase in AMPA receptor density and premature maturation of excitatory synapses (*Rumbaugh et al., 2006*; *Clement et al., 2012*; *Clement et al., 2013*). Unexpectedly, here we found that *Syngap1* haploinsufficiency restricted to MGE-derived interneurons depresses glutamatergic synaptic transmission, potentially via presynaptic mechanisms. In addition, LIV PV+ cells receive the strongest thalamocortical input compared to excitatory cells and other subpopulations of GABAergic interneurons (*Ji et al., 2016*; *Zurita et al., 2018*). Our study indicates that a reduction in AMPA-mediated thalamocortical transmission onto LIV PV+ cells may contribute to the deficit in glutamatergic drive, as suggested by facilitated PPR in cHet compared to control mice. This, combined with the increased onset latencies of thalamocortical-evoked AMPA responses along with an enhanced NMDA/AMPA ratio, points to both a likely decrease in presynaptic release from thalamocortical fibers and impaired recruitment of PV+ cells by thalamic inputs in cHet mice.

How could *Syngap1* haploinsufficiency in Nkx2.1-expressing cells affect the glutamatergic drive coming from local and thalamic excitatory cells? Our data suggest a role of Syngap1 in promoting GABAergic cell-intrinsic excitability during normal development. These findings are in line with recent data reporting a decrease in intrinsic excitability of developing cortical excitatory cells in *Syngap1*+/− mice (*Arora et al., 2022*). In PV+ cells, connectivity and cell excitability are reciprocally regulated at the circuit level (*Favuzzi et al., 2017*); thus, it is possible that early-onset *Syngap1* haploinsufficiency in MGE-derived interneurons may first affect the development of their intrinsic excitability properties, which in turn would modulate the maturation of their excitatory drive and activity levels in adulthood. Alternatively, homeostatic adaptation of PV+ interneurons in response to the decreased number of excitatory inputs could trigger changes in voltage-gated D-type K+ currents (*Dehorter et al., 2015*; *Favuzzi et al., 2017*). Therefore, Syngap1 may play complex roles at the cellular level and at different developmental stages, based on the crosstalk between neuronal activity levels and Syngap1 localization and interaction with other proteins. The use of a conditional genetic strategy to induce *Syngap1* haploinsufficiency specifically in MGE-derived interneurons allows investigating its effects in these GABAergic populations; however, it is important to highlight the limit of this approach, since whether GABAergic interneurons physiology would be similarly affected within an entire network carrying the same mutation (thus affecting also excitatory neurons) remains to be established.

Based on morphology and synaptic targets, three main subgroups of PV+ cells have been identified in auditory cortex, i.e., basket cells (BCs), chandelier cells, and long-range projecting PV+ cells (*Levy and Reyes, 2012*; *Rock et al., 2018*; *Zurita et al., 2018*; *Bertero et al., 2019*). Despite differences

in morphology and target selectivity, different PV+ cell subgroups share common electrophysiological features such as short AP-half width, low input resistance, very high threshold current, and relatively small AP amplitude. In particular, while clusters of atypical BC PV+ have been previously reported in several areas (*Helm et al., 2013*; *Nassar et al., 2015*; *Bengtsson Gonzales et al., 2020*; *Ekins et al., 2020*), auditory cortex PV+ BCs have been considered as a homogeneous electrophysiological group (*Studer and Barkat, 2022*). One of the key findings of this study is that mature A1 contains at least two distinct morphological and electrophysiological subgroups of PV+ BCs, including a subgroup with an unexpectedly broader AP half-width (*Supplementary file 1*). The genetic tools used to identify PV+ cells (*PV_Cre* or G42 mice vs *Nkx2.1*_Cre mice) might selectively label a specific PV+ BC cell subtype. Alternatively, the distribution of different PV+ BC subtypes could depend on the cortical region and layer. The presence of at least two PV+ BC subtypes with different electrophysiological characteristics may partly explain the previously observed variability in the recruitment of auditory cortex PV+ cells in vivo (*Seybold et al., 2015*; *Phillips and Hasenstaub, 2016*; *Keller et al., 2018*; *Gothner et al., 2021*), as these differences could arise from the AP half-width criteria used to sort FS cells.

Despite the differences observed in how *Syngap1* haploinsufficiency affects the anatomy and physiological properties of the two PV+ cell subtypes, a shared deficit we observed was the decrease in intrinsic excitability, as suggested by the increased AP threshold affecting AP initiation. In PV+ cells, voltage-gated D-type K$^+$ currents mediated by the Kv1 family strongly contribute to AP generation, making them effective targets for modifying AP latency, threshold, and rheobase current in these cells (*Wang et al., 1994*; *Goldberg et al., 2008*; *Zurita et al., 2018*). Here, we indeed restored the excitability of mutant PV+ cells by pharmacologically inhibiting this channel family using α-DTX; however, whether Kv1 currents or/and channel density is altered in mutant PV+ cells remains to be investigated. Consistent with our findings, *Arora et al., 2022* rescued pyramidal cell-intrinsic excitability and neuronal morphology via lowering elevated potassium channels, by expressing a dominant negative form of the Kv4.2 potassium channel subunit, dnKv4.2, in developing *Syngap1* mutant mice. Kv4.2 and Kv1 potassium channels modulate the intrinsic excitability of pyramidal cell and PV+ interneurons, respectively, indicating that the action of Syngap1 on potassium channels may be a general mechanism. A recent study focusing on the PSD interactomes of Syngap1 isolated from adult homogenized mouse cortex suggested a physical interaction between Syngap1 and the potassium channel auxiliary subunit Kvβ2 (*Wilkinson et al., 2017*). Since Kvβ2 regulates the translocation of Kv1 channels in dopaminergic neurons and potentially PV+ cells (*Okaty et al., 2009*; *Yee et al., 2022*), it is possible that *Syngap1* haploinsufficiency may lead to dysregulated Kv1 translocation at the membrane, leading to excessive K currents. Of note, both Kv1 channels and Syngap1 are developmentally regulated (*Okaty et al., 2009*; *Gamache et al., 2020*), thus *Syngap1* hypofunction may lead to dysregulation of genes encoding voltage-gated potassium channel affecting cell maturation and excitability.

Interestingly, the alterations in intrinsic excitability were less pronounced in cHet SST+ interneurons, primarily affecting their firing rate. This discrepancy could be due to the heterogeneity of SST+ interneurons likely present in our dataset (*Scala et al., 2019*; *Hostetler et al., 2023*). In addition, it's possible that other intrinsic factors, not assessed in this study, may have contributed to this effect. For example, in SST+ cells the differences in the AHP kinetics depend predominantly on the presence of a second slower AHP component impacting the overall amplitude, slope, and duration of the AHP (*Riedemann et al., 2018*). Recent studies also showed that SYNGAP1 interacts with Kv4 (*Wilkinson et al., 2017*). Since somato-dendritic Kv4 channels in SST+ interneurons contribute to the regulation of their firing (*Serôdio and Rudy, 1998*; *Bourdeau et al., 2007*), *Syngap1* haploinsufficiency could affect SST+ cell excitability via this channel.

PV+ cells have a fundamental role in generating and maintaining gamma oscillations in the brain (*Cardin et al., 2009*). In particular, recent studies have also shown that deficiency in PV interneuron-mediated inhibition contributes to increased baseline cortical gamma rhythm (*Spencer, 2011*; *Carlén et al., 2012*), a phenotype we observed in the auditory cortex of germline *Syngap1$^{+/-}$* mutant mice, SYNGAP1-ID patients (*Carreño-Muñoz et al., 2022*) and mice with conditional *Syngap1* haploinsufficiency restricted to MGE-derived interneurons (*Jadhav et al., 2024*). Specifically, PV+ interneurons, which target the perisomatic domain of pyramidal neurons, are adapted for fast synchronization of network activity controlling spike timing of the excitatory network in auditory cortex (*Wehr and Zador, 2003*; *Li et al., 2014*). To sharpen the tuning of neighboring pyramidal cells, PV+ interneurons need to be more effectively recruited by excitatory inputs so that they can restrict the temporal summation of

excitatory responses of their pyramidal cell targets and increase the temporal precision of their firing (*Povysheva et al., 2006*). Our study suggests that a decrease in AMPA-mediated thalamocortical input onto LIV PV+ cells along with deficits in their intrinsic excitability, could account for altered spike timing of pyramidal cells, causing an increase in overall network excitability.

An open question remains whether synaptic properties differ among PV+ cell subtypes, which we could not address due to technical limitations. These include the lack of specific neurochemical markers to distinguish between the two PV+ subtypes (*Ekins et al., 2020*), and the use of a Cs$^+$-based internal solution required for voltage-clamp experiments, which prevents the recording of neuronal firing. It could be interesting to correlate PV expression levels directly with AP half-width, since BC-short may express higher levels of PV compared to BC-broad as already found in the striatum using patch seq approach (*Bengtsson Gonzales et al., 2020*). Further, in our studies, we used a conditional mouse model where *Syngap1* haploinsufficiency is restricted to specific cell types, namely MGE-derived interneurons. Since cell-type-specific genetic mutations do not typically occur in humans, it would be interesting to investigate whether *SYNGAP1*-haploinsufficient human-derived neurons show alterations in specific GABAergic subpopulations – intrinsic and synaptic properties. Further, whether and how PV+ physiology is affected in global haploinsufficient mice remains to be addressed. Of note, global haploinsufficient *Syngap1* mice, *SYNGAP1*-ID patients, and MGE-restricted *Syngap1* haploinsufficient mice show comparable abnormal phenotypes in cortical auditory processing (*Carreño-Muñoz et al., 2022*; *Jadhav et al., 2024*). Further experiments specifically targeting PV+ cell activity, using targeted chemogenetic or pharmacological approach (*Kourdougli et al., 2023*) are required to shed light on the role PV+ interneuron hypoactivity in these phenotypes.

## Materials and methods
### Mice
All procedures and experiments were done in accordance with the Comité Institutionnel de Bonnes Pratiques Animales en Recherche (CIBPAR) of the CHU Ste-Justine Research Center in line with the principles published in the Canadian Council on Animal's Care's (protocol reference number: 2024-6519). Mice were housed (2–5 per cage), maintained in a 12/12 hr light/dark cycle, and given ad libitum access to food and water. Experiments were performed in 9- to 13-week-old male mice during the light phase. To investigate the effects of *Syngap1* haploinsufficiency in cortical PV+ and SST+ interneurons, we generated mice heterozygous for the *Syngap1* conditional allele (*Syngap1$^{f/f}$*; Jackson Laboratories; #029303, RRID:IMSR_JAX:029303) under the Tg(*Nkx2.1-Cre*) driver line (Jackson Laboratories; #008661, RRID:IMSR_JAX:008661) and further crossed them with mice carrying the *Rosa26$^{LSL-EGFP}$* reporter allele (indicated as RCE in Results; Jackson Laboratories; #032037, RRID:MMRRC_032037-JAX) to generate Tg(*Nkx2.1-Cre*):*Rosa26$^{LSL-EGFP f/f}$*:*Syngap1$^{+/+}$* and Tg(*Nkx2.1-Cre*):*Rosa26$^{LSL-EGFP f/f}$*:*Syngap1$^{f/+}$* mice, for targeted recordings. Nkx2.1-expressing cells were identified by the expression of EGFP, since the *Rosa26$^{LSL-EGFP}$* allele allows Cre-dependent EGFP expression in MGE-derived interneurons.

### Acute slice preparation
Briefly, animals (age range, mean ± SEM: 75.5 ± 1.8 postnatal days for control group and 72.1 ± 1.7 postnatal days in cHet group) were anesthetized deeply with ketamine–xylazine (ketamine: 100 mg/kg, xylazine: 10 mg/kg), transcardially perfused with 25 ml of ice-cold cutting solution (containing the following in mM: 250 sucrose, 2 KCl, 1.25 NaH$_2$PO$_4$, 26 NaHCO$_3$, 7 MgSO$_4$, 0.5 CaCl$_2$, and 10 glucose, pH 7.4, 330–340 mOsm/l) and decapitated. The brain was then dissected carefully and transferred rapidly into an ice-cold (0–4°C) cutting solution. Auditory thalamocortical slices (thickness, 350 μm) containing A1 and the medial geniculate nucleus were prepared. For A1 slices, the cutting angle was 15° from the horizontal plane (lateral raised; *Cruikshank et al., 2002*; *Zhao et al., 2009*; *Meng et al., 2017*). Auditory thalamocortical slices were cut in the previously mentioned ice-cold solution using a vibratome (VT1000S; Leica Microsystems or Microm; Fisher Scientific) and transferred to a heated (37.5°C) oxygenated recovery solution containing the following (in mM): 124 NaCl, 2.5 KCl, 1.25 NaH$_2$PO$_4$, 26 NaHCO$_3$, 3 MgSO$_4$, 1 CaCl$_2$, and 10 glucose; pH 7.4; 300 mOsm/l, and allowed to recover for 45 min. Subsequently, during experiments, slices were continuously perfused (2 ml/min) with standard artificial cerebrospinal fluid (ACSF) containing the following (in mM): 124 NaCl, 2.5 KCl,

1.25 NaH$_2$PO$_4$, 26 NaHCO$_3$, 2 MgSO$_4$, 2 CaCl$_2$, and 10 glucose, pH 7.4 saturated with 95% O$_2$ and 5% CO$_2$ at near physiological temperature (30–33°C). We did not observe any difference between control and cHet mice in terms of slice quality, success rate of recordings, and cellular health.

## Whole-cell patch-clamp recording

PV+ and SST+ neurons located in LIV of A1 cortex were visually identified as EGFP-expressing somata under an epifluorescence microscope with blue light (filter set: 450–490 nm). All electrophysiological recordings were carried out using a 40× water-immersion objective. Recording pipettes were pulled from borosilicate glass (World Precision Instruments) with a PP-83 two-stage puller (Narishige) to a resistance range of 5–7 MΩ when backfilled with intracellular solution. Whole-cell patch-clamp recordings from PV+ and SST+ interneurons were performed in voltage or current-clamp mode. Pipette capacitance was neutralized, and bridge balance applied. For voltage-clamp recording, we used an intracellular Cs$^+$-based solution containing (in mM): 130 CsMeSO$_4$, 5 CsCl, 2 MgCl$_2$, 10 phosphocreatine, 10 HEPES, 0.5 EGTA, 4 ATP-TRIS, 0.4 GTP-TRIS, 0.3% biocytin, and 2 QX-314 (pH 7.2–7.3; 280–290 mOsm/l). These recordings were performed to analyze the excitatory drive received by PV+ and SST+ cells. Series resistance in voltage-clamp was monitored throughout the experiment, and cells that had substantial changes in series resistance (>15%) during recording were discarded. The reported voltage values were not compensated for the junction potential. Recordings of sEPSC and mEPSCs were performed in voltage-clamp at –70 mV in the presence of gabazine (1 µM; Tocris Bioscience) and CGP55845 (2 µM; Abcam Biochemicals). sEPSCs were first sampled over a 2-min period. A TTX (1 µM; Alomone Labs) containing perfusion solution was then added (flow rate of 2 ml/min) and, after a 5-min interval, mEPSCs were sampled over a 2-min period. Recordings of spontaneous and miniature inhibitory postsynaptic currents (sIPSC, mIPSCs) were performed in voltage-clamp at +10 mV in the presence of NBQX (10 µM; Abcam Biochemicals) and DL-AP5 (100 µM; Abcam Biochemicals) for sISPCs and with addition of TTX (1 µM) for mIPSCs. For recordings of thalamocortical electrically evoked AMPA/NMDA EPSC in LIV PV+ cells, current pulses (2-ms duration, 25–1000 µA) were delivered to the thalamic radiation every 30 s via a tungsten concentric bipolar microelectrode placed in the white matter midway between the medial geniculate nucleus and the A1 (rostral to the hippocampus). Electrical stimulation of thalamic radiation may activate not only monosynaptic thalamic fibers but also polysynaptic (corticothalamic and/or corticocortical) EPSC component. To identify monosynaptic thalamo-cortical connections, we used as criteria the onset latencies of EPSC and the variability jitter obtained from the standard deviation of onset latencies. Onset latencies were defined as the time interval between the beginning of the stimulation artifact and the onset of the EPSC. Monosynaptic connections are characterized by short onset latencies and low jitter variability (*Richardson et al., 2009*; *Blundon et al., 2011*; *Chun et al., 2013*). In our experiments, the initial slopes of EPSCs evoked by white matter stimulation had short onset latencies (mean onset latency, 4.27 ± 0.11 ms, N = 16 neurons in controls, and 5.07 ± 0.07 ms, N=14 neurons in cHet mice) and low onset latency variability jitter (0.24 ± 0.03 ms in controls vs 0.31 ± 0.03 ms in cHet mice), suggestive of activation of monosynaptic thalamocortical monosynaptic connections (*Richardson et al., 2009*; *Blundon et al., 2011*; *Chun et al., 2013*). Evoked AMPA (eAMPA) currents were recorded at –70 mV in the presence of CGP (2 µM) and gabazine (1 µM), while evoked NMDA (eNMDA) currents were recorded at +40 mV in the presence of NBQX (10 µM) and confirmed afterward with the application of DL-AP5 (100 µM). For PPR experiments, local synaptic stimulation in LIV was achieved using a bipolar stimulating electrode made from borosilicate theta-glass capillaries (BT-150-10, Sutter Instruments, Novato, CA) filled with ACSF. Two EPSCs were evoked in LIV BC with two electric pulses (0.2 ms, 40–140 µA each) at five intervals (30, 50, 100, 200, and 500 ms) every 10 s in the presence of picrotoxin (100 µM; Tocris/Cedarlane).

Current-clamp recordings were obtained in ACSF containing synaptic blockers gabazine (1 µM), CGP55845 (2 µM), and kynurenic acid (2 mM). For these recordings, we used an intracellular K$^+$-based solution containing (in mM): 130 KMeSO$_4$, 2 MgCl$_2$, 10 di-Na-phosphocreatine, 10 HEPES, 4 ATP-Tris, 0.4 GTP-Tris, and 0.3% biocytin (Sigma), pH 7.2–7.3, 280–290 mOsm/l. Passive and active membrane properties were analyzed in current clamp mode: active membrane properties were recorded by subjecting cells to multiple current step injections (step size 40 pA) of varying amplitudes (–200 to 600 pA). In subsequent experiments, the same protocol was repeated in the presence of the voltage-gated potassium channel (Kv) blocker α-DTX (100 nM; Alomone labs). In these experiments, slices

were recovered in a holding chamber for at least 1 hr in the presence of α-DTX. Once placed in the recording chamber, slices were kept for recording for a maximum of 1 hr. Passive membrane properties, resting membrane potential (Vm), input resistance (Rin), and membrane capacitance (Cm) were obtained immediately after membrane rupture. Membrane potentials were maintained at −80 mV, series resistances (10–18 MΩ) and Rin were monitored on-line with a 40 pA current injection (150 ms) given before each 500 ms current injection stimulus. Only cells with resting membrane potential more negative than −60 mV at the start of recording and spikes with overshoot were considered for further analysis. We performed bridge balance and neutralized the capacitance before starting every recording. The bridge balance was monitored throughout the experiment, and neurons showing changes of >15% in bridge balance during the recording were discarded. Data acquisition (filtered at 2–3 kHz and digitized at 10 kHz; Digidata 1440, Molecular Devices, CA, United States) was performed using the Multiclamp 700B amplifier and the Clampex 10.6 software (Molecular Devices).

## Electrophysiological data analysis

All analysis was performed by researchers blind to the genotype. Analysis of electrophysiological recordings was performed using Clampfit 10.7 (Molecular Devices, RRID:SCR_011323). For the analysis of sEPSCs, mEPSCs, sIPSCs, and mIPSCs, a minimum of 100 events were sampled per cell over a 2 min period using an automated template search algorithm in Clampfit. The 20–80% rise time of the response and the decay time constant determined from the exponential fit (100–37%) were calculated. Charge transfer was calculated by integrating the area under the EPSC and IPSC waveforms. The mean PSC synaptic current was calculated as the charge transfer of the averaged PSC ($\Delta Q$) multiplied by mean PSC frequency. For thalamocortical eAMPA and eNMDA, the mean amplitude of EPSCs including both failure and success was obtained from a total of 5–10 sweeps. Onset latency indicated the time from beginning of stimulus artifact and the onset of eAMPA or eNMDA. To measure eNMDA/eAMPA ratios, the eAMPA component was taken at the peak of EPSC at −70 mV, whereas the eNMDA component was measured at the peak of EPSC at +40 mV. For PPR experiments, the PPR was calculated as the ratio between the mean AUC of the second response and the mean AUC of the first response (7–10 sweeps).

For the analysis of current-clamp recordings from PV cells, threshold current was measured as the minimal current necessary to evoke an AP. For the analysis of the AP properties, the first AP appearing within a 50-ms time window from the beginning of the current pulse was analyzed. AP latency was measured as the time between current step onset and when membrane voltage reached AP threshold. The AP amplitude was measured from the AP threshold to the peak. The AP half-width was measured at the voltage level of the half of AP amplitude. The AP rise and fall time were measured between the AP threshold and the maximal AP amplitude, and between the maximal AP amplitude and the AP end, respectively. The fAHP amplitude was determined as the minimum voltage following the AP peak subtracted from the AP threshold. fAHP time was determined as the time between AP threshold and the negative peak of fAHP. The hyperpolarization-activated cation current ($I_h$)-associated voltage rectification ($I_h$ sag) was determined as the amplitude of the membrane potential sag from the peak hyperpolarized level to the level at the end of the hyperpolarizing step when the cell was hyperpolarized to −100 mV. Membrane time constant ($\tau$) was calculated by the product of $R_{in}$ and $C_m$. For the firing analysis, we considered only APs with amplitude >30 mV as full APs. Spikelets with amplitude smaller than 30 mV were not analyzed in this study. The inter-spike interval (ISI) was determined by the time difference between adjacent AP peaks. Spike amplitude accommodation ratio was calculated by dividing the amplitude of the last AP by the amplitude of the first generated in response to 2× threshold current injection. Firing frequency adaptation ratios were calculated by dividing the last ISI with the first one of the responses to 2× threshold current injection. The maximal (initial) firing frequency ($F_{max}$initial) was computed as the reciprocal of the first ISI in a spike train elicited by the current step (max +600 pA) applied before a noticeable appearance of spikelets. The steady-state firing frequency ($F_{ss}$) was computed as the reciprocal of the average of the last four ISIs in the spike train where $F_{max}$initial was obtained. Finally, the number of APs (# APs)–current relationship for evoked firing was determined by injecting 500 ms somatic current steps of increasing amplitude (40 pA increments) to a maximum of 600 pA. For current-clamp recordings in the presence of α-DTX experiments, the delta ($\Delta$) values were calculated for threshold current, AP threshold, and AP number at +200 pA, by subtracting individual values of α-DTX-treated cells from the average of their respective control group.

## Hierarchical clustering and PCA

This analysis was based on previously published data finding heterogeneity in PV+ interneurons population (*Helm et al., 2013*), and performed using the software IBM SPSS V29.0.0 RRID:SCR_016479. Hierarchical clustering was based on Euclidean distance of PV+ cells from control mice. To identify potential clusters, we used AP half-width and $F_{max}$initial values. We then performed multidimensional cluster analysis on passive and active membrane properties to identify possible common groupings of PV+ interneurons using the software Prism 9.0 (GraphPad Software, RRID:SCR_002798). In our database, we focused on 13 parameters (see *Figure 5b*) that were for the majority unrelated. *Figure 5b* is a cross-correlation matrix of these 13 parameters with correlation indices shade-coded. Black means perfectly positively correlated (correlation index of 1.0), white means perfectly negatively correlated (correlation index of –1.0), and light gray means not correlated (correlation index of 0). In our database, we have parameters that are not strongly correlated (e.g., threshold current and AP amplitude, fAHP amplitude and AP latency), and others that are correlated (correlation coefficient >0.5 or <–0.5; AP latency and amplitude AR; AP half-width and AP amplitude; Rin and AP half-width; AP amplitude and $F_{max}$ initial). We retained all parameters because they encompass different features of membrane properties (*Helm et al., 2013*). We therefore performed PCA on the 13 parameters to reduce the dimensionality and to potentiate cluster separation.

## Immunohistochemistry, cell reconstruction, and anatomical identification

For post-hoc anatomical identification, every recorded neuron was filled with biocytin (0.5%, Sigma) during whole-cell recordings (15 min). To reveal biocytin, the slices were permeabilized with 0.3% Triton-X 100 and incubated at 4°C with a streptavidin-conjugated Alexa-488 (1:1000, Molecular Probes, Cat# S32354, RRID:AB_2315383) in TBS. For PV and SST immunofluorescence, sections were permeabilized with 0.25% Triton-X 100 in PBS and incubated in blocking solution containing 20% normal goat serum (NGS) for 1 hr. Then, sections were incubated with the following primary antibodies diluted in 1% NGS, 0.25% Triton-X 100 in PBS: were incubated with primary antibodies mouse anti-PV (1:1000, Swant, Cat# 235, RRID:AB_10000343) and rabbit anti-SST (1:1000, Thermofisher Invitrogen, Cat# PA5-82678, RRID:AB_2789834) at 4°C for 48–72 hr. Sections were then washed in PBS (3 × 10 min each), incubated for 2 hr at RT with the following secondary antibodies diluted in 1% NGS, 0.25% Triton-X 100 in PBS and mounted on microscope slides: Alexa 555-conjugated goat anti-rabbit (1:1000; Life Technologies, A21430, RRID:AB_2535851) and Alexa 647-conjugated goat anti-mouse (1:250, Cell Signaling, 4410S, RRID:AB_1904023). Whole biocytin-filled cells were acquired with a 1 µm step using a 20× objective (NA 0.75) on a Leica SP8-DLS confocal microscope. For each imaged neuron, we noted the spatial distribution across cortical layers of the axonal arbor and whether its dendrites carried spines. Confocal stacks of PV+ neurons were merged for detailed reconstruction in Neuromantic tracing software version 1.7.5 (RRID:SCR_013597, *Myatt et al., 2012*). Dendritic arbors were reconstructed plane-by-plane from the image z-stack and analyzed using the Neuromantic software. All the reconstructions used for dendritic analysis contained intact cells with no evident cut dendrites. Sholl analysis of reconstructed dendritic arbors was performed in FIJI software (RRID:SCR_002285) using the plugin Neuroanatomy. This analysis was performed in radial coordinates, using a 10-µm step size from *r*=0, with the origin centered on the cell soma, and counting the number of compartments crossing a given radius. All analysis was performed by researchers blind to the genotype.

## vGlut1/PSD95 and vGlut 2/PSD immunostaining, imaging, and quantification

P60 mice were anesthetized with: ketamine-100 mg/kg + xylazine-10 mg/kg + acepromazine-10 mg/kg and perfused transcardially with 0.9% saline followed by 4% paraformaldehyde (PFA) in phosphate buffer (0.1 M PB, pH 7.2–7.4). Brains were dissected out and post-fixed in 4% PFA overnight at 4°C. They were subsequently transferred to 30% sucrose (prepared in PBS, pH 7.2) at 4°C for 48 hr. Brains were then embedded in molds filled with OCT Tissue Tek and frozen in a bath of 2-methybutane placed on a bed of dry ice and ethanol. Coronal sections were cut at 40 µm with a cryostat (Leica CM3050 S) and collected as floating sections in PBS. Brain sections were first permeabilized in 0.2% Triton-X in PBS for 1 hr at RT and then blocked in 10% normal donkey serum (NDS) with 0.2% Triton-X

100 and 5% bovine serum albumin (BSA) in PBS for 2 hr at RT followed by incubation at 4°C for 48 hr with the following primary antibodies diluted in 5% NDS, 0.2% Triton-X 100 and 2% BSA in PBS: goat anti-PV (1:1000, Swant, Cat# PVG-213, RRID:AB_2721207), rabbit anti-VGlut1 (1:100 Thermo Fisher/Invitrogen, Cat# 48-2400, RRID:AB_2533843), rabbit anti-VGlut2 1:1000, Synaptic systems, Cat# 135402, RRID:AB_2187539 mouse anti-PSD95 (1:500, Invitrogen, Cat# 6G6-1C9, RRID:AB_2092361). Sections were then washed in PBS +0.1% Triton-X 100 (3 × 10 min each) and incubated for 2 hr at RT with the following secondary antibodies diluted in 5% NDS, 0.2% Triton-X 100 and 2% BSA in PBS: Alexa 488-conjugated donkey anti-rabbit (1:500, Life technologies/Invitrogen, Cat# A11055, RRID:AB_2534102), Alexa 555-conjugated donkey anti-mouse (1:500, Life technologies/Invitrogen, Cat# A31570, RRID:AB_2536180), Alexa 633-conjugated donkey anti-goat (1:500, Invitrogen, Cat# A21082, RRID:AB_2535739). Sections were rinsed in 0.1% Triton-X 100 in PBS (3 × 10' each + 1 × 5') and mounted with Vectorshield mounting medium (Vector).

Immunostained sections were imaged using a Leica SP8-STED confocal microscope, with an oil immersion 63× (NA 1.4) at 1024 × 1024, zoom = 1, z-step = 0.3 μm, stack size of ~15 μm. Images were acquired from the A1 from at least three coronal sections per animal. All the confocal parameters were maintained constant throughout the acquisition of an experiment. All images shown in *Figure 1—figure supplement 3a and c* are from a single confocal plane. To quantify the number of vGlut1/PSD95 or vGlut2/PSD95 putative synapses, images were exported as TIFF files and analyzed using Fiji (Image J) software. We first manually outlined the profile of each PV cell soma (identified by PV immunolabeling). At least four innervated somata were selected in each confocal stack. We then used a series of custom-made macros in Fiji as previously described (*Chehrazi et al., 2023*). After subtracting background (rolling value = 10) and Gaussian blur ($\sigma$ value = 2) filters, the stacks were binarized and vGlut1/PSD95 or vGlut2/PSD95 puncta were independently identified around the perimeter of a targeted soma in the focal plane with the highest soma circumference. Puncta were quantified after filtering particles for size (included between 0 and 2 μm$^2$) and circularity (included between 0 and 1). Quantification of the density of perisomatic puncta colocalizing both VGlut1-PSD95 and VGlut2-PSD95 was normalized to controls. All analysis was performed by researchers blind to the genotype. No mouse was excluded from this analysis.

## Statistics

Data were expressed as mean ± SEM. For statistical analysis, we based our conclusion on the statistical results generated by LMM, modeling animal as a random effect and genotype as fixed effect. We used this statistical analysis because we considered the number of mice as independent replicates and the number of cells in each mouse as repeated measures (*Berryer et al., 2016*; *Heggland et al., 2019*; *Yu et al., 2022*). For cumulative distributions, the same number of events was chosen randomly from each cell and analyzed by LMM, modeling animal as a random effect and genotype as fixed effect. Two-way ANOVA with Sidak's multiple comparison post hoc test was used for the detection of differences in AP firing and number of dendritic intersections between genotypes. Experimenters were blind to genotypes during recording, neuron reconstruction, and data analysis. All experiments were done using a minimum of six mice of at least three to four cohorts from different mouse litters. Statistical analysis was performed using Sigma Plot 11.0 (RRID:SCR_003210), Prism 9.0 (GraphPad Software, RRID:SCR_002798) and IBM SPSS V29.0.0 (RRID:SCR_016479) for LMM analysis.

## Acknowledgements

We are very grateful to Dr. Lisa Topolnik for her invaluable suggestions and help. We would like to thank James Bellord Waldron for his technical assistance, the Comité Institutionnel de Bonne Pratiques Animales en Recherche (CIBPAR), all the personnel of the animal facility of the Research Center of CHU Sainte-Justine (Université de Montreal), Compute Canada, and the Plateforme Imagerie Microscopique (PIM) of the Research Center of CHU Sainte-Justine for their instrumental technical support and all lab members for insightful data discussion. This work was supported by the Canadian Institutes of Health Research (GDC, SK), Natural Sciences and Engineering Research Council of Canada (SK), Rare Diseases: Model and Mechanisms Network (GDC), Jonathan-Bouchard Chair in intellectual disability (JLM) and Fonds UdeM pour le partenariat CHU Sainte-Justine -Institut Imagine en épilepsie de l'enfant (GDC, JLM). RF and JLDK are supported by Fonds de Recherche du Québec en Santé (FRQS), and Savoy Foundation fellowship.

# Additional information

### Funding

| Funder | Grant reference number | Author |
|---|---|---|
| Canadian Institutes of Health Research | | Graziella Di Cristo Said Kourrich |
| Fonds de Recherche du Québec-Santé | | Ruggiero Francavilla |
| Fonds de Recherche Québec-Santé | | Jorelle Linda Damo Kamda |
| Natural Sciences and Engineering Research Council of Canada | | Graziella Di Cristo |
| Savoy Foundation | | Ruggiero Francavilla |

The funders had no role in study design, data collection, and interpretation, or the decision to submit the work for publication.

### Author contributions

Ruggiero Francavilla, Conceptualization, Data curation, Formal analysis, Investigation, Visualization, Writing - original draft, Writing - review and editing; Bidisha Chattopadhyaya, Conceptualization, Formal analysis, Supervision, Validation, Investigation, Visualization, Writing - review and editing; Jorelle Linda Damo Kamda, Data curation, Formal analysis, Investigation, Visualization; Vidya Jadhav, Data curation, Formal analysis, Investigation, Visualization, Writing - review and editing; Said Kourrich, Supervision, Validation, Methodology, Writing - original draft, Writing - review and editing; Jacques L Michaud, Conceptualization, Resources, Funding acquisition, Writing - review and editing; Graziella Di Cristo, Conceptualization, Supervision, Funding acquisition, Validation, Writing - original draft, Writing - review and editing

### Author ORCIDs

Ruggiero Francavilla ⓘ http://orcid.org/0009-0004-0192-6646
Jorelle Linda Damo Kamda ⓘ https://orcid.org/0000-0001-7140-4334
Graziella Di Cristo ⓘ https://orcid.org/0000-0003-4464-4994

### Ethics

All procedures and experiments were done in accordance with the Comité Institutionnel de Bonnes Pratiques Animales en Recherche (CIBPAR) of the CHU Ste-Justine Research Center in line with the principles published in the Canadian Council on Animal's Care's (protocol reference number: 2024-6519).

Reviewer #2 (Public review): https://doi.org/10.7554/eLife.97100.4.sa1
Author response https://doi.org/10.7554/eLife.97100.4.sa2

---

# Additional files

### Supplementary files

MDAR checklist

Source data 1. Membrane properties of BC-broad vs BC-short in control mice-raw data.

Source data 2. sEPSCs in SST+ cells from control vs SST+ cells from cHet mice-raw data.

Supplementary file 1. Membrane properties of BC-broad vs BC-short in control mice-Table.

Supplementary file 2. sEPSCs in SST+ cells from control vs SST+ cells from cHet mice-Table.

### Data availability

All data generated or analyzed during this study are included in the manuscript and supporting files.

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
