## [Editor Report · eLife Assessment]

This study provides **valuable** evidence indicating that SynGap1 regulates the synaptic drive and membrane excitability of parvalbumin- and somatostatin-positive interneurons in the auditory cortex. Since haplo-insufficiency of SynGap1 has been linked to intellectual disabilities without a well-defined underlying cause, the central question of this study is timely. The experimental data is **solid**, as in their revisions the authors successfully addressed questions related to changes in thalamocortical presynaptic excitability, the contradiction between spontaneous and mini EPSCs data, and the anatomical analysis of excitatory synapses.

---

## [Referee Report · Reviewer #2 (Public review)]

In this manuscript, the authors investigated how partial loss of SynGap1 affects inhibitory neurons derived from the MGE in the auditory cortex, focusing on their synaptic inputs and excitability. While haplo-insufficiently of SynGap1 is known to lead to intellectual disabilities, the underlying mechanisms remain unclear.

This is the third revision of the manuscript that has improved further, and the main issues were addressed. Specifically, the Authors addressed the contradiction of mEPSC and sEPSC data of the previous version by new experiments and revision of the manuscript text. While alternative explanations are still possible, the new control experiments provide necessary background for reproducibility and the manuscript text puts the observations in the right context. Furthermore, the manuscript now appropriately emphasizes that anatomical analysis was restricted to somatic excitatory synapses. Thus, the readers will be aware of the potential limitations of these measurements.

Strengths:

The questions are novel and relevant. Most of the issues in the experimental design are solved or answered.

Weaknesses:

Despite the interesting and novel questions, there are potential alternative interpretations of the observations, but these cannot be addressed within the breadth of a single paper.

---

## [Author Response]

The following is the authors’ response to the previous reviews

**Reviewer #2 (Public review):**
Summary:In this manuscript, the authors investigated how partial loss of SynGap1 affects inhibitory neurons derived from the MGE in the auditory cortex, focusing on their synaptic inputs and excitability. While haplo-insufficiently of SynGap1 is known to lead to intellectual disabilities, the underlying mechanisms remain unclear.Strengths:The questions are novelWeaknesses:Despite the interesting and novel questions, there are significant issues regarding the experimental design and potential misinterpretations of key findings. Consequently, the manuscript contributes little to our understanding of SynGap1 loss mechanisms.Major issues in the second version of the manuscript:In the review of the first version there were major issues and contradictions with the sEPSC and mEPSC data, and were not resolved after the revision, and the new control experiments rather confirmed the contradiction.In the original review I stated: "One major concern is the inconsistency and confusion in the intermediate conclusions drawn from the results. For instance, while the sEPSC data indicates decreased amplitude in PV+ and SOM+ cells in cHet animals, the frequency of events remains unchanged. In contrast, the mEPSC data shows no change in amplitudes in PV+ cells, but a significant decrease in event frequency. The authors conclude that the former observation implies decreased excitability. However, traditionally, such observations on mEPSC parameters are considered indicative of presynaptic mechanisms rather than changes of network activity. The subsequent synapse counting experiments align more closely with the traditional conclusions. This issue can be resolved by rephrasing the text. However, it would remain unexplained why the sEPSC frequency shows no significant difference. If the majority of sEPSC events were indeed mediated by spiking (which is blocked by TTX), the average amplitudes and frequency of mEPSCs should be substantially lower than those of sEPSCs. Yet, they fall within a very similar range, suggesting that most sEPSCs may actually be independent of action potentials. But if that was indeed the case, the changes of purported sEPSC and mEPSC results should have been similar." Contradictions remained after the revision of the manuscript. On one hand, the authors claimed in the revised version that "We found no difference in mEPSC amplitude between the two genotypes (Fig. 1g), indicating that the observed difference in sEPSC amplitude (Figure 1b) could arise from decreased network excitability". On the other hand, later they show "no significative difference in either amplitude or inter-event intervals between sEPSC and mEPSC, suggesting that in acute slices from adult A1, most sEPSCs may actually be AP independent." The latter means that sEPSCs and mEPSCs are the same type of events, which should have the same sensitivity to manipulations.

We thank the reviewer for the detailed comments. Our results suggest a diverse population of PV+ cells, with varying reliance on action potential-dependent and -independent release. Several PV+ cells indeed show TTX sensitivity (reduced EPSC event amplitudes following TTX application: See new Supplementary Figure 2b-e), but their individual responses are diluted when all cells are pooled together. To account for this variability, we recorded sEPSC followed by mEPSC from more mice of both genotypes (new Figure 1f-j). Further, following the editors and reviewers’ suggestions, we removed speculations about the role of network activity changes.

In summary, our data confirmed that TTX blocked APs in PV+ cells and that recordings were stable as indicated by lack of changes in series resistance during the recording period in our experimental setup (new Suppl. Figure 2f-i). We found no difference in mEPSC amplitude between the two genotypes (Fig. 1g, right), indicating that the observed difference in sEPSC amplitude (Figure 1c, right) could be due to impaired AP-dependent release in cHet mice and the presence of large-amplitude sEPSCs that are preferentially affected by TTX in control mice (new Suppl. Figure 2b-e). Conversely, cHet mice showed longer inter-mEPSC time interval (cumulative distribution in Figure 1g, left), and significantly lower charge transfer and DQ*f (Figure 1j) compared to controls littermates, suggesting a decrease of glutamatergic presynaptic release sites onto PV+ cells.

Concerns about the quality of the synapse counting experiments were addressed by showing additional images in a different and explaining quantification. However, the admitted restriction of the analysis of excitatory synapses to the somatic region represent a limitation, as they include only a small fraction of the total excitation - even if, the slightly larger amplitudes of their EPSPs are considered.

We agree with the reviewer that restricting the anatomical analysis of excitatory synapses to PV cell somatic region is a limitation, as highlighted it in the discussion of the revised manuscript. Recent studies, based on serial block-face scanning electron microscopy, suggest that cortical PV+ interneurons receive more robust excitatory inputs to their perisomatic region as compared to pyramidal neurons (see for example, Hwang et al. 2021, Cerebral Cortex, http://doi.org/10.1093/cercor/bhaa378). It is thus possible that putative glutamatergic synapses, analysed by vGlut1/PSD95 colocalisation around PV+ cell somata, may be representative of a substantially major excitatory input population. Since analysing putative excitatory synapses onto PV+ dendrites would be difficult and require a much longer time, we re-phrased the text to more clearly highlight the rationale and limitation of this approach.

New experiments using paired-pulse stimulation provided an answer to issues 3 and 4. Note that the numbering of the Figures in the responses and manuscript are not consistent.

We are glad that the reviewer found that the new paired-pulse experiments answered previously raised concerns. We corrected the discrepancy in figure numbers in the manuscript. Thank you for noticing.

I agree that low sampling rate of the APs does not change the observed large differences in AP threshold, however, the phase plots are still inconsistent in a sense that there appears to be an offset, as all values are shifted to more depolarized membrane potentials, including threshold, AP peak, AHP peak. This consistent shift may be due to a non-biological differences in the two sets of recordings, and, importantly, it may negate the interpretation of the I/f curves results (Fig. 5e).

We agree with the reviewers that higher sampling rate would allow to more accurately assess different parameters, such as AP peak, half-width, rise time, etc., while it would not affect the large differences in AP threshold we observed between control and mutant mice. Since the phase plots to not add to our result analysis, we removed them from the revised manuscript.

Additional issues:The first paragraph of the Results mentioned that the recorded cells were identified by immunolabelling and axonal localization. However, neither the Results nor the Methods mention the criteria and levels of measurements of axonal arborization.

Recorded MGE-derived interneurons were filled with biocytin, and their identity was confirmed by immunolabeling for neurochemical markers (PV or SST) and analysis of anatomical properties. In particular, whole biocytin-positive immunolabelled neurons were acquired using a Leica SP8-DLS confocal microscope (20x objective, NA 0.75; Z-step 1 1μm). For each imaged neuron, which was the result of multiple merged confocal stacks, we visually determined the spatial distribution across cortical layers of the axonal arbor and whether its dendrites carried spines. We added this information in the method section. Furthermore, to better represent our methodological approach, we added a new figure (Supplemental Figure 1) including (1) two examples of PV+ interneurons, showing dendrites devoid of spines and axons spreading from layer II to layer V (new Suppl. Figure 1a); and (2) two examples of SST+ interneurons showing dendritic with spines and axons projecting from layer IV to layer I where they gave rise to multiple collaterals (new Suppl. Figure 1b).

The other issues of the first review were adequately addressed by the Authors and the manuscript improved by these changes.

We are happy the reviewer found that the other issues were well addressed.

**Reviewer #3 (Public review):**
This paper compares the synaptic and membrane properties of two main subtypes of interneurons (PV+, SST+) in the auditory cortex of control mice vs mutants with Syngap1 haploinsufficiency. The authors find differences between control and mutants in both interneuron populations, although they claim a predominance in PV+ cells. These results suggest that altered PVinterneuron functions in the auditory cortex may contribute to the network dysfunctions observed in Syngap1 haploinsufficiency-related intellectual disability.The subject of the work is interesting, and most of the approach is rather direct and straightforward, which are strengths. There are also some methodological weaknesses and interpretative issues that reduce the impact of the paper.(1) Supplementary Figure 3: recording and data analysis. The data of Supplementary Figure 3 show no differences either in the frequency or amplitude of synaptic events recorded from the same cell in control (sEPSCs) vs TTX (mEPSCs). This suggests that, under the experimental conditions of the paper, sEPSCs are AP-independent quantal events. However, I am concerned by the high variability of the individual results included in the Figure. Indeed, several datapoints show dramatically different frequencies in control vs TTX, which may be explained by unstable recording conditions. It would be important to present these data as time course plots, so that stability can be evaluated. Also, the claim of lack of effect of TTX should be corroborated by positive control experiments verifying that TTX is working (block of action potentials, for example). Lastly, it is not clear whether the application of TTX was consistent in time and duration in all the experiments and the paper does not clarify what time window was used for quantification.

We understand the reviewer’s concern about high variability. To account for this variability, we recorded sEPSC followed by mEPSC from more mice of both genotypes (see new Figure 1f-j). We confirmed that TTX worked as expected several times through the time course of this study, in different aliquots prepared from the same TTX vial that was used for all experiments. The results of the last test we performed, showing that TTX application blocks action potentials in a PV+ cell, are depicted in new Suppl. Figure 2a. Furthermore, new Suppl. Figure 2f-i shows series resistance (Rs) over time for 4 different PV+ interneurons, indicating recording stability. These results are representative of the entire population of recorded neurons, which we have meticulously analysed one by one. TTX was applied using the same protocol for all recorded neurons. In particular, sEPSCs were first sampled over a 2 min period. A TTX (1μM; Alomone Labs)-containing solution was then perfused into the recording chamber at a flow rate of 2 mL/min. We then waited for 5 min before sampling mEPSCs over a 2 min period. We added this information in the revised manuscript methods.

(2) Figure 1 and Supplementary Figure 3: apparent inconsistency. If, as the authors claim, TTX does not affect sEPSCs (either in the control or mutant genotype, Supplementary Figure 3 and point 1 above), then comparing sEPSC and mEPSC in control vs mutants should yield identical results. In contrast, Figure 1 reports a _selective_ reduction of sEPSCs amplitude (not in mEPSCs) in mutants, which is difficult to understand. The proposed explanation relying on different pools of synaptic vesicles mediating sEPSCs and mEPSCs does not clarify things. If this was the case, wouldn't it also imply a decrease of event frequency following TTX addition? However, this is not observed in Supplementary Figure 3. My understanding is that, according to this explanation, recordings in control solution would reflect the impact of two separate pools of vesicles, whereas, in the presence of TTX, only one pool would be available for release. Therefore, TTX should cause a decrease in the frequency of the recorded events, which is not what is observed in Supplementary Figure 3.

To account for the large variability and clarify these results, we recorded sEPSCs followed by mEPSCs from more mice of both genotypes (new Figure 1f-j). We found no difference in mEPSC amplitude between the two genotypes (Fig. 1g, right), indicating that the observed difference in sEPSC amplitude (Figure 1c, right) could be due to impaired AP-dependent release in cHet mice and the presence of large-amplitude sEPSCs that are preferentially affected by TTX in control mice (new Suppl. Figure 2b-e). Conversely, cHet mice showed longer inter-mEPSC time interval (cumulative distribution in Figure 1g, left), and significantly lower charge transfer and DQ*f (Figure 1j) compared to controls littermates, suggesting a decrease of glutamatergic presynaptic release sites. We rephrased the text in the revised manuscript according to the updated data and, following the reviewer’s suggestions, we removed speculations relying on different pools of synaptic vesicles.

(3) Figure 1: statistical analysis. Although I do appreciate the efforts of the authors to illustrate both cumulative distributions and plunger plots with individual data, I am confused by how the cumulative distributions of Figure 1b (sEPSC amplitude) may support statistically significant differences between genotypes, but this is not the case for the cumulative distributions of Figure 1g (inter mEPSC interval), where the curves appear even more separated. A difference in mEPSC frequency would also be consistent with the data of Supplementary Fig 2b, which otherwise are difficult to reconciliate. I would encourage the authors to use the Kolmogorov-Smirnov rather than a t-test for the comparison of cumulative distributions.

We thank the reviewer for this thoughtful suggestion. We recorded more mice of both genotypes and the updated data now show a significant difference between the cumulative distributions of the inter mEPSC intervals recorded from the two genotypes (new Figure 1g). For statistical analysis, we based our conclusion on the statistical results generated by LMM, modelling animal as a random effect and genotype as fixed effect. We used this statistical analysis because we considered the number of mice as independent replicates and the number of cells in each mouse as repeated measures (Berryer et al. 2016; Heggland et al., 2019; Yu et al., 2022). For cumulative distributions, the same number of events was chosen randomly from each cell and analysed by LMM, modelling animal as a random effect and genotype as fixed effect. The reason we decided to use LMM for our statistical analyses is based on the growing concern over reproducibility in biomedical research and the ongoing discussion on how data are analysed (see for example, Yu et al (2022), Neuron 110:21-35 https://doi.org/10.1016/j.neuron.2021.10.030; Aarts et al. (2014). Nat Neurosci **17**, 491–496. https://doi.org/10.1038/nn.3648). We acknowledge that patch-clamp data has been historically analysed using t-test and analysis of variance (ANOVA), or equivalent nonparametric tests. However, these tests assume that individual observations (recorded neurons in this case) are independent of each other. Whether neurons from the same mouse are independent or correlated variables is an unresolved question, but does not appear to be likely from a biological point of view. Statisticians have developed effective methods to analyze correlated data, including LMM.

(4) Methods. I still maintain that a threshold at around -20/-15 mV for the first action potential of a train seems too depolarized (see some datapoints of Fig 5c and Fig7c) for a healthy spike. This suggest that some cells were either in precarious conditions or that the capacitance of the electrode was not compensated properly.

As suggested by the reviewer, in the revised figures we excluded the neurons with threshold at -20/-15 mV. In addition, we performed statistical analysis with and without these cells (data reported below) and found that whether these cells are included or excluded, the statistical significance of the results does not change.

Fig.5c: including the 2 outliers from cHet group with values of -16.5 and 20.6 mV: 42.6±1.01 mV in control, n=33 cells from 15 mice vs -35.3±1.2 mV in cHet, n=40 cells from 17 mice, ***p<0.001, LMM; excluding the 2 outliers from cHet group -42.6±1.01 mV in control, n=33 cells from 15 mice vs -36.2±1.1 mV in cHet, n=38 cells from 17 mice, ***p<0.001, LMM.

Fig.7c: including the 2 outliers from cHet group with values of -16.5 and 20.6 mV: 43.4±1.6 mV in control, n=12 cells from 9 mice vs -33.9±1.8 mV in cHet, n=24 cells from 13 mice, **p=0.002, LMM; excluding the 2 outliers from cHet group -43.4±1.6 mV in control, n=12 cells from 9 mice vs -35.4±1.7 mV in cHet, n=22 cells from 13 mice, *p=0.037, LMM.

(5) The authors claim that "cHet SST+ cells showed no significant changes in active and passive membrane properties (Figure 8d,e); however, their evoked firing properties were affected with fewer AP generated in response to the same depolarizing current injection".This sentence is intrinsically contradictory. Action potentials triggered by current injections are dependent on the integration of passive and active properties. If the curves of Figure 8f are different between genotypes, then some passive and/or active property MUST have changed. It is an unescapable conclusion. The general _blanket_ statement of the authors that there are no significant changes in active and passive properties is in direct contradiction with the current/#AP plot.

We agreed with the reviewer and rephrased the abstract, results and discussion according to better represent the data. As discussed in the previous revision, it's possible that other intrinsic factors, not assessed in this study, may have contributed to the effect shown in the current/#AP plot.

(6) The phase plots of Figs 5c, 7c, and 7h suggest that the frequency of acquisition/filtering of current-clamp signals was not appropriate for fast waveforms such as spikes. The first two papers indicated by the authors in their rebuttal (Golomb et al., 2007; Stevens et al., 2021) did not perform a phase plot analysis (like those included in the manuscript). The last work quoted in the rebuttal (Zhang et al., 2023) did perform phase plot analysis, but data were digitized at a frequency of 20KHz (not 10KHz as incorrectly indicated by the authors) and filtered at 10 kHz (not 2-3 kHz as by the authors in the manuscript). To me, this remains a concern.

We agree with the reviewer that higher sampling rate would allow to more accurately assess different AP parameters, such as AP peak, half-width, rise time, etc. The papers were cited in context of determining AP threshold, not performing phase plot analysis. We apologize for the confusion and error. Finally, we removed the phase plots since they did not add relevant information.

(7) The general logical flow of the manuscript could be improved. For example, Fig 4 seems to indicate no morphological differences in the dendritic trees of control vs mutant PV cells, but this conclusion is then rejected by Fig 6. Maybe Fig 4 is not necessary. Regarding Fig 6, did the authors check the integrity of the entire dendritic structure of the cells analyzed (i.e. no dendrites were cut in the slice)? This is critical as the dendritic geometry may affect the firing properties of neurons (Mainen and Sejnowski, Nature, 1996).

As suggested by the reviewer, we removed Fig.4. All the reconstructions used for dendritic analysis contained intact cells with no evidently cut dendrites.